

**FRONT MATTER**
**Title**
Mid-Holocene Antarctic sea-ice increase driven by marine ice sheet retreat
**Authors**
Kate E. Ashley[1*], James A. Bendle[1], Robert McKay[2], Johan Etourneau[3], Francis J. Jimenez-Espejo[3,4],
Alan Condron[5], Anna Albot[2], Xavier Crosta[6], Christina Riesselman[7,8], Osamu Seki[9], Guillaume Massé[10],
Nicholas R. Golledge[2,11], Edward Gasson[12], Daniel P. Lowry[2], Nicholas E. Barrand[1], Katelyn Johnson[2],
Nancy Bertler[2], Carlota Escutia[3] and Robert Dunbar[13].
**Affiliations**
[1]School of Geography, Earth and Environmental Sciences, University of Birmingham, Edgbaston,
Birmingham, B15 2TT, UK
[2]Antarctic Research Centre, Victoria University of Wellington, Wellington 6140, New Zealand
[3]Instituto Andaluz de Ciencias de la Tierra (CSIC), Avenida de las Palmeras 4, 18100 Armilla, Granada,
Spain
[4]Department of Biogeochemistry, Japan Agency for Marine-Earth Science and Technology
(JAMSTEC), Yokosuka 237-0061, Japan
[5]Department of Geology and Geophysics, Woods Hole Oceanographic Institution, Woods Hole, MA
02543, USA
[6]UMR-CNRS 5805 EPOC, Université de Bordeaux, 33615 Pessac, France
[7]Department of Geology, University of Otago, Dunedin 9016, New Zealand
[8]Department of Marine Science, University of Otago, Dunedin 9016, New Zealand
[9]Institite of Low Temperature Science, Hokkaido University, Sapporo, Hokkaido, Japan
[10]TAKUVIK, UMI 3376 UL/CNRS, Université Laval, 1045 avenue de la Médecine, Quebec City,
Quebec, Canada G1V 0A6
[11]GNS Science, Avalon, Lower Hutt 5011, New Zealand
[12]Department of Geography, University of Sheffield, Winter Street, Sheffield, S10 2TN, UK
[13]Department of Environmental Earth Systems Science, Stanford University, Stanford, A 94305-2115
[*]**Corresponding Author:** email: ken155@student.bham.ac.uk





## 1. ABSTRACT

Over recent decades Antarctic sea-ice extent has increased, alongside widespread ice shelf thinning and freshening of waters along the Antarctic margin. In contrast, Earth system models generally simulate a decrease in sea ice. Circulation of water masses beneath large cavity ice shelves is not included in current models and may be a driver of this phenomena. We examine a Holocene sediment core off East Antarctica that records the Neoglacial transition, the last major baseline shift of Antarctic sea-ice, and part of a late-Holocene global cooling trend. We provide a multi-proxy record of Holocene glacial meltwater input, sediment transport and sea-ice variability. Our record, supported by high-resolution ocean modelling, shows that a rapid Antarctic sea-ice increase occurred against a backdrop of increasing glacial meltwater input and gradual climate warming. We suggest that mid-Holocene ice shelf cavity expansion led to supercooling of surface waters and sea-ice growth which slowed basal ice shelf melting. Incorporating this feedback mechanism into global climate models will be important for future projections of Antarctic changes.

## 2. INTRODUCTION

Ice shelves and sea ice are intrinsically linked and represent fundamental components of the global climate system, impacting ice-sheet dynamics, large-scale ocean circulation, and the Southern Ocean biosphere. Antarctic ice-shelves with large sub-shelf cavities (e.g. Ross, Filchner-Ronne) play a key role in regional sea-ice variations, by cooling and freshening surface ocean waters for hundreds of kilometres beyond the ice shelf edge (Hellmer, 2004; Hughes *et al.*, 2014). Antarctic sea ice has expanded over the past few decades, particularly in the western Ross Sea region (Turner *et al.*, 2016), alongside widespread thinning of ice shelves (Paolo *et al.*, 2015) and freshening along the Antarctic margin (Jacobs *et al.*, 2002; Aoki *et al.*, 2013). The drivers and feedbacks involved in these decadal trends are still poorly understood, hampered by the sparse and short-term nature of meteorological, oceanographic and glaciological observations (Jones *et al.*, 2016), and thus establishing the long-term trajectory for Antarctic sea ice on the background of accelerated ice sheet loss remains a challenge. Marine sediment cores provide a longer-term perspective and highlight a major baseline shift in coastal sea ice ~4.5 ka ago (Steig *et al.*, 1998; Crosta *et al.*, 2008; Denis *et al.*, 2010) which characterizes the mid-Holocene 'Neoglacial' transition in the Antarctic. A mechanistic driver for this climate shift currently remains





unresolved, but we propose that two interrelated aspects of the last deglaciation are significantly
underrepresented in current models of this transition: (i) the retreat of grounded ice sheets from the
continental shelves of Antarctica, and (ii) the subsequent development of large ice shelf cavities during
the Holocene. Both factors would significantly alter water mass formation on Antarctica's continental
shelves, which today are major source regions of Antarctic Bottom Water (AABW) and Antarctic
Surface Water (AASW). These interrelated processes are underrepresented in coupled ocean-atmosphere
models which currently do not simulate the timing, magnitude and rapid onset of the Neoglacial
(Supplementary Materials).

Integrated Ocean Drilling Program (IODP) Expedition 318 cored a 171 m thick deposit of laminated
diatomaceous ooze at Site U1357 offshore Adélie Land (Fig. 1), deposited over the past 11,400 years.
Here, we present a new Holocene record of glacial meltwater, sedimentary input and local sea ice
concentrations from Site U1357 using compound-specific hydrogen isotopes of fatty acid biomarkers
($\delta^2 H_{FA}$), terrigenous grain size, biogenic silica accumulation, highly-branched isoprenoid alkenes (HBIs)
and Ba/Ti ratios (Fig. 2 and S4).

We interpret $\delta^2 H_{FA}$ (Fig. 2) fluctuations in Adélie Drift sediments as a record of meltwater input from
isotopically-depleted glacial ice (Supplementary Materials). Antarctic glacial ice is highly depleted in $^2H$
compared to ocean water, thus creating highly contrasting end-member values for the two major H
source pools. Grain size (sand and mud percentage and sorting), natural gamma radiation (NGR) and
terrigenous and biosiliceous mass accumulation rates (MARs) reflect changing sediment delivery either
driven via local glacial meltwater discharge or advection of suspended sediment by oceanic currents.
The diene/triene HBI ratio is used as a proxy for coastal sea ice presence (Massé *et al.*, 2011). Ba/Ti
enrichment is considered to reflect enhanced primary productivity. These records allow a unique
opportunity to reconstruct the magnitude of the coupled response of the ocean and ice sheet during the
Neoglacial transition. Details on all proxies and associated uncertainties can be found in Section 4.2 and
in the Supplementary Materials.


**3. MATERIALS AND METHODS**





### 3.1 Organic geochemical analyses

Lipid extraction of sediment samples was performed at the Royal Netherlands Institute for Sea Research (NIOZ). Freeze-dried and homogenized samples were extracted by Dionex™ accelerated solvent extraction (DIONEX ASE 200) using a mixture of dichloromethane (DCM)/methanol (MeOH) (9:1, v/v) at a temperature of 100°C and a pressure of $7.6 \times 10^6$ Pa (Kim *et al.*, 2010).

Two-thirds of the total lipid extract were sent to the University of Glasgow, UK and separated over an aminopropyl silica gel column where the total neutral fraction was eluted with 4ml of DCM/ isopropanol (1:1 v/v), and the total acid fractions were eluted into an 8ml vial with 4% acetic acid in ethyl-ether solution (Huang *et al.*, 1999). Derivatisation to Fatty Acid Methyl Esters was achieved by adding 200 μl of MeOH containing 14% v/v Boron triflouride to the 8ml vial containing the TAF. The vial was seal and placed in the drying cabinet at 70°C for one hour. The MeOH was dried under $N_2$ and the FAMES were recovered and cleaned up by eluting through a pre-cleaned 3cm silica gel column (60 A; 35-70) with 4ml of hexane and 4ml of DCM (containing the FAMES). The recovered FAMES fraction was split 50:50 for compound specific carbon and hydrogen analysis, respectively. $\delta^2H$ values indicate depletion against the international standards: Vienna Pee Dee Belemnite (V-PDB) is the standard for $\delta13C$ and Vienna Standard Mean Ocean Water (V-SMOW) for $\delta^2H$.

Compound specific hydrogen isotope analyses of FAMES was performed at the Institute of Low Temperature Science, Hokkaido University. $\delta^2H$ values were obtained using a CS-IRMS system with a HP 6890 gas chromatograph and a ThermoQuest Finnigan MAT Delta Plus XL mass spectrometer. Separation of the FAMES was achieved with a HP-5 MS fused silica capillary column (30 m x 0.32 mm i.d., film thickness of 0.25 μm) with a cooled on-column injector. An *n*-alkane and a reference gas whose isotopic values were known was co-injected with the samples as an internal isotopic standard for $\delta^2H$. A laboratory standard (Mix F8 of FAMES from Indiana University) containing $C_{10}$–$C_{30}$ FAMES was analyzed daily to check the accuracy and the drift of the instrument and to normalize the data to the SMOW/SLAP isotopic scale. The $H^{3+}$ factor was measured every three days.

### 3.2 Inorganic geochemical analysis and electronic microscopy

Major element concentrations were obtained using X-Ray Fluorescence Scanner on 412 analyses measured directly over undisturbed sediment sections. The bulk major element composition included in





this study was measured between sections U1357B-1H-2 to U1357-19H-5 continuously each 50 cm. We
used an Avaatech X-ray fluorescence (XRF-Scanner) core scanner at the IODP-Core Repository/Texas
A&M University laboratories (USA) during December 2010. Non-destructive XRF core-scanning
measurements were performed over 1 cm$^2$ area with slit size of 10 mm, a current of 0.8 mA and
sampling time of 45 seconds at 10 kV in order to measure the relative content of titanium (Ti) and
barium (Ba).

Field emission scanning electron microscopy (FESEM) images and corresponding spectrum were
obtained with an AURIGA FIB-FESEM Carl Zeiss SMT at Centro de Instrumentación Científica,
Granada University, Spain

**3.3 Grain size analyses**
A total of 341 samples were prepared for grain size analysis. Samples were treated for removal of
biogenic opal with a 1M sodium hydroxide NaOH solution and incubated in a water bath at 80°C for 24
hours. This procedure was repeated twice due to an incomplete dissolution of diatoms observed in smear
slides. The samples were then treated with $H_2O_2$ to remove organic material at 80°C for 24 hours.
Samples were measured using a Beckman Coulter LS 13 320 Laser Diffraction Particle Size Analyser
(LPSA). The LPSA has a relatively narrow range of optimum obscuration which is determined by the
sample surface area, in turn determined by sample concentration and sample distribution. Prior to grain
size analysis, ~30 mL of 0.5 g/L Calgon (sodium hexametaphosphate) was added to the samples, and
sonicated and stirred in order to disperse the grains and prevent clumping. The range in sample mass for
most of the post-treatment samples varied from ~0.035-0.8 grams. Random biases propagating through
this process cannot be ruled out, especially when taking account of susceptibility of grains <10 μm to
clump(McCave *et al.*, 1995) and random cohesion of grains due to any remaining organic content. The
aqueous liquid module in the LPSA also does not accurately record the <2 μm clay that may
compromise a significant part of the size spectrum in glacial environments (McCave *et al.*, 1995;
McCave and Hall, 2006). Given these considerations, subsamples were taken from a total of 84 samples
to test reproducibility of the data relating to sub-sampling biases, with a least squares regression
showing a high reproducibility with an r$^2$ value of 0.744. An additional 12 samples were sub-sampled
before the chemical treatment in order to test the reproducibility of the treatment methodology, with a
least squares regression showing a high reproducibility with an r$^2$ value of 0.752.




### 3.4 HBIs


Highly branched isoprenoids (HBI) alkenes were extracted at Laboratoire d'Océanographie et du
Climat: Experimentations et Approches Numériques (LOCEAN), separately from the fatty acids, using a
mixture of 9mL $CH_2Cl_2$/MeOH (2:1, v:v) to which internal standards were added and applying several
sonication and centrifugation steps in order to extract properly the selected compounds(Etourneau *et al.*,
2013). After drying with $N_2$ at 35°C, the total lipid extract was fractionated over a silica column into an
apolar and a polar fraction using 3 mL hexane and 6 mL $CH_2Cl_2$/MeOH (1:1, v:v), respectively. HBIs
were obtained from the apolar fraction by the fractionation over a silica column using hexane as eluent
following the procedures reported by Belt *et al.* (2007; Massé *et al.*, 2011). After removing the solvent
with $N_2$ at 35°C, elemental sulfur was removed using the TBA (Tetrabutylammonium) sulfite method
(Jensen *et al.*, 1977; Riis and Babel, 1999). The obtained hydrocarbon fraction was analyzed within an
Agilent 7890A gas chomatograph (GC) fitted with 30 m fused silica Agilent J&C GC column (0.25 mm
i.d., 0.25 µm film thickness), coupled to an Agilent 5975C Series mass selective detector (MSD).
Spectra were collected using the Agilent MS-Chemstation software. Individual HBIs were identified on
the basis of comparison between their GC retention times and mass spectra with those of previously
authenticated HBIs (Johns *et al.*, 1999) using the Mass Hunter software. Values are expressed as
concentration relative to the internal standard.

### 3.5 Biogenic silica


Biogenic silica concentrations (wt% BSi) were measured on 349 discrete samples using a molybdate
blue spectrophotometric method modified from (Strickland and Parsons, 1970; DeMaster, 1981).
Analytical runs included replicates from the previous sample group and from within the run, and each
run was controlled by 10 standards and a blank with dissolved silica concentrations ranging from 0 µM
to 1200 µM. For each analysis, ~7 mg of dry, homogenized sediment was leached in 0.1M NaOH at
85°C, and sequential aliquots were collected after 2, 3, and 4 hours. Following addition of reagents,
absorbance of the 812 nm wavelength was measured using a Shimadzu UV-1800 spectrophotometer.
Dissolved silica concentration of each unknown was calculated using the standard curve, and data from
the three sampling hours were regressed following the method of DeMaster (1981) to calculate wt%
BSi. In our U1357B samples, wt% BSi ranges from maximum of ~60% in early and mid-Holocene light





laminae to a minimum of 31% in late Holocene dark laminae. The average standard deviation of
replicate measurements is 0.5%.

**3.6 Model simulations**
All numerical calculations were performed using the Massachusetts Institute of Technology general
circulation model (MITgcm) (Marshall *et al.*, 1997); a three-dimensional, ocean sea-ice, hydrostatic,
primitive equation model. The experiments presented here were integrated on a global domain projected
onto a cube-sphere grid to permit a relatively even grid spacing and to avoid polar singularities (Adcroft
*et al.*, 2004; Condron and Winsor, 2012). The ocean grid had a mean, eddy-permitting, horizontal grid
spacing of 1/6° (18-km) with 50 vertical levels ranging in thickness from 10m near the surface to
approximately 450m at the maximum model depth. The ocean model is coupled to a sea-ice model in
which ice motion is driven by forces generated by the wind, ocean, Coriolis force, and surface elevation
of the ocean, while internal ice stresses are calculated using a viscous-plastic (VP) rheology, as
described in Zhang and Hibler (1997). In all experiments, the numerical model is configured to simulate
present-day (modern) conditions: Atmospheric forcings (wind, radiation, rain, humidity etc.) are
prescribed using 6-hourly climatological (1979-2000) data from the ERA-40 reanalysis product
produced by the European Centre for Medium-range Weather Forecasts and background rates of runoff
from the ice sheet to the ocean are based on the numerical ice sheet model of Pollard and Deconto
(2016) integrated over the same period (1979-2000). To study the pathway of meltwater in the ocean,
additional fresh (i.e. 0 psu) water was released into the surface layer of the ocean model at the grid
points closest to the front of the Ross Ice Shelf. Five different discharge experiments were performed by
releasing meltwater into this region at rates of 0.01 Sv (Sv = $10^6$ m3/s), 0.05 Sv, 0.1 Sv, 0.5 Sv, and 1 Sv
for the entire duration of each experiment (~3.5 years).

4.    ENVIRONMENTAL SETTING AND INTERPRETATION OF PROXY DATA
We utilize a 180 m thick sediment core that was recovered from the Wilkes Land Margin continental
shelf in the Adélie Basin (IODP Site U1357). This core targeted an expanded sediment drift (Adélie
Drift) and provides a high-resolution Holocene record of climate variability. Below we provide pertinent
details on this unique site and on our application of compound specific $\delta^2$H measurements on algal
biomarkers as a novel meltwater proxy. Further details on proxy interpretation (Ba/Ti, grain size, HBIs)
are given in the Supplementary Materials.



### 4.1 The Adélie Drift

Site U1357 is located in the Dumont d'Urville Trough of the Adélie Basin, ca. 35 km offshore from Adélie Land (66°24.7990'S, 140°25.5705'E; Fig 1). This is a >1000 m deep, glacially scoured depression on the East Antarctic continental shelf, bounded to the east by the Adélie Bank. Further east lays the Adélie Depression and the Mertz Bank, the latter located north of the Mertz Glacier floating ice tongue. The Adélie Land region is dissected by several glaciers which could potentially contribute terrigenous sediment into the coastal zone with the core site located 40 km to the north of the Astrolabe Glacier, and ca. 75 and 300 km northwest of the Zélée and Mertz glaciers, respectively.

The site itself is located within the Dumont d'Urville polynya (DDUP), which has a summer (winter) extent of 13,020 km$^2$ (920 km$^2$), but is also directly downwind and downcurrent of the much larger and highly productive Mertz Glacier polynya (MGP) to the east, with a summer (winter) extent of 26,600 km$^2$ (591 km$^2$) (Arrigo and van Dijken, 2003). The MGP forms as a result of reduced sea-ice westward advection due to the presence of the Mertz Glacier Tongue (Massom *et al.*, 2001) and strong katabatic winds which blow off the Antarctic ice sheet with temperatures below -30°C (Bindoff *et al.*, 2000). Katabatic winds freeze the surface waters and blow newly formed ice away from the coast, making the polynya an efficient sea-ice 'factory', with higher rates of sea-ice formation in comparison to non-polynya ocean areas which undergo seasonal sea ice formation (Kusahara *et al.*, 2010). The MGP produces 1.3% of the total Southern Ocean sea ice volume despite occupying less than 0.1% of total Antarctic sea ice extent (Marsland *et al.*, 2004).

As a result of the upwelling polynya environments, the area along the Adélie Coast is characterized by extremely high primary productivity, with the water column known to host significant amounts of phytoplankton, dominated by diatoms (Beans *et al.*, 2008). The Mertz Glacier zone is generally characterized by stratified waters in the summer, due to seasonal ice melt, with these conditions corresponding to the highest phytoplankton biomass. The lack of ice cover means polynyas are the first polar marine systems exposed to spring solar radiation, making them regions of enhanced biological productivity compared to adjacent waters. A considerable amount of resultant sedimentation is focused via the westward flowing currents from both of these polynyas within the deep, protected Adélie Basin,





resulting in a remarkably high sedimentation rate of ca. 1.5-2 cm year$^{-1}$ at Site U1357 (Escutia *et al.*,

245     2011).


Although biogenic and terrigenous sediment is interpreted to be sourced locally in the Adélie Land
region, the mass accumulation rate of these sediments in this drift is associated with the intensity of
westward flowing currents (S2.2). Critically, these westward currents also act to transport water masses
from further afield, and Site U1357 is directly oceanographically downstream of the Ross Sea, meaning
the continental shelf in this region receives significant Antarctic Surface Water (ASSW) transported by
the Polar Easterlies and the Antarctic Slope Current (ASC) from the Ross Sea embayment (Fig 3). Thus,
changes in the surface waters of the Ross Sea influence Site U1357. Whitworth et al. (1998) confirm the
continuity of the westward flowing ASC between the Ross Sea and the Wilkes Land margin. This flow
is largely associated with the Antarctic Slope Front, which reflects the strong density contrast between
AASW and Circumpolar Deep Water (CDW). McCartney and Donohue (2007) estimate that the
transport in the westward ASC, which links the Ross Sea to the Wilkes Land margin, reaches 76 Sv (106
m$^3$ s$^{-1}$). This contributes to a cyclonic gyre, which together with the ASC dominate the circulation at Site
U1357. The gyre transports around 35 Sv, and comes mainly from the Ross Sea region, with a lesser
contribution from a westward flow associated with the Antarctic Circumpolar Current.
While several small glaciers within Adélie Land may contribute meltwater to the site, the region is also
likely to be influenced significantly by changes in Ross Sea waters. Freshwater release simulations from
the Ross Ice Shelf (RIS) confirm this oceanographic continuity between the Ross Sea and the Wilkes
region (Fig 3). Five simulations with fluxes from 0.01 to 1 Sv released from the edge of the RIS all
indicate that meltwater is almost completely entrained within the westward coastal surface current and
reaches Site U1357 within 4 months to 1 year (Fig 4). These fluxes cover a wide range of meltwater
inputs and show a strong linear relationship with salinity at the core site (Fig. 4). This suggests that the
magnitude of the signal recorded at Site U1357 is directly related to the magnitude of meltwater.
Local processes do also play a critical role in this region. For example, episodic calving events of the
Mertz Glacier tongue release fast ice over the drill site and create strong surface water stratification,
cutting off local AABW production (Campagne *et al.*, 2015). Although appearing to be only a local
process, there is still a regional (Ross Sea) influence, as this fast ice that builds up behind the Mertz
Glacier is formed by the freezing of fresher AASW transported from the Ross Sea (Fig 3). Thus,





conditions in the Ross Sea, such as the melting of isotopically depleted glacial ice, would influence both
the isotopic composition and amount of this sea ice.

**4.2 Site specific interpretation of $\delta^2 H_{FA}$ as a glacial meltwater proxy**

**4.2.1. Source of fatty acids**
To best interpret the hydrogen isotope signal recorded by the $C_{18}$ FA, it is important to determine the
most likely source these compounds are derived from, and thus the habitat in which they are produced.
The $C_{18}$ FA, however, is known to be produced by a wide range of organisms and so we cannot preclude
the possibility of multiple sources, especially in a highly diverse and productive region such as the
surface waters of offshore Adélie Land. However, we can attempt to determine the most dominant
producer(s), which will help us understand the main signal being recorded by the isotopes.

An analysis of the FAs within eight classes of microalgae by Dalsgaard *et al.* (2003) (compiling results
from multiple studies) showed *Cryptophyceae, Chlorophyceae, Prasinophyceae* and *Prymnesiophyceae*
to be the most dominant producers of total $C_{18}$ FAs. The *Bacillariophyceae* class, on the other hand,
which includes the diatoms, were found to produce only minor amounts of $C_{18}$ FA, instead synthesizing
abundant $C_{16:1}$ FAs. Thus, despite the water column offshore Adélie Land being dominated by diatoms,
these are unlikely to be a major source of the $C_{18}$ FA within U1357B (Beans *et al.*, 2008; Riaux-Gobin
*et al.*, 2011).

Of the four microalgae classes dominating $C_{18}$ production (Dalsgaard *et al.*, 2003), species from the
Chlorophyceae and Prymnesiophyceae classes have been observed within surface waters offshore
Adélie Land after spring sea-ice break-up (Riaux-Gobin *et al.*, 2011). Here, *Phaeocystis antarctica* of
the Prymnesiophytes was found to dominate the surface water phytoplankton community (representing
16% of the phytoplankton assemblage), whereas Cryptophyceae spp. were found in only minor
abundances (Riaux-Gobin *et al.*, 2011). In the Antarctic, *Phaeocystis* is thought to be the most dominant
producer of $C_{18}$ FAs (Dalsgaard *et al.*, 2003), and thus is likely to be a key producer of the $C_{18}$ FA in
U1357B samples.



To investigate this further, we measured compound-specific carbon isotopes of the $C_{18}$ FAs in U1357B
samples, which gives an average $\delta^{13}C$ value of -29.8 ± 1.0 ‰ (n=85). Budge *et al.* (2008) measured a
similar $\delta^{13}C$ value of -30.7 ± 0.8‰ from $C_{16}$ FAs derived from Arctic pelagic phytoplankton, while sea
ice algae and higher trophic level organisms all had much higher $\delta^{13}C$ values (sea ice algae having
values of -24.0 ± 2.4‰). Assuming similar values apply for the $C_{18}$ FA and for organisms within the
water column at our site, this suggests that our $C_{18}$ FA is predominantly derived from pelagic
phytoplankton.

Furthermore, $\delta^{13}C$ measurements of suspended particulate organic matter (SPOM) near Prydz Bay, East
Antarctica by Kopczynska *et al.* (1995) showed that sites with high $\delta^{13}C$ SPOM values (-20.1 to -
22.4‰) were characterized by diatoms and large heterotrophic dinoflagellates, whereas the lowest $\delta^{13}C$
SPOM values (-29.7 to -31.85‰) were associated with *Phaeocystis*, naked flagellates and autotrophic
dinoflagellates. Wong and Sackett (1978) measured the carbon isotope fractionation of seventeen
species of marine phytoplankton and showed that Haptophyceae (of which *Phaeocystis* belongs) had the
largest fractionation of -35.5‰.

Therefore, based on the known producers of $C_{18}$ FAs, observed phytoplankton assemblages within
modern surface waters offshore Adélie Land, and the $\delta^{13}C$ value of $C_{18}$ FAs in U1357B samples, as
discussed above, we argue that the $C_{18}$ FA here is predominantly produced by *Phaeocystis* (most likely
*P. antarctica*), but with potential minor inputs from other algal species such as Cryptophytes or diatoms.

*Phaeocystis antarctica* is a major phytoplankton species within the Antarctic, dominating spring
phytoplankton blooms, particularly in the Ross Sea (DiTullio *et al.*, 2000; Schoemann *et al.*, 2005). It is
known to exist both within sea ice and in open water (Tang *et al.*, 2008) and has been observed in
surface waters in great abundance following spring sea-ice break-up, at both coastal and offshore sites in
Adélie Land (Riaux-Gobin *et al.*, 2011).

Although a large proportion of organic matter produced in the surface water is recycled in the upper
water column, the small fraction which is deposited in the sediment reaches the sea floor through large
particles sinking from above as "marine snow". This export production includes large algal cells, fecal
pellets, zooplankton carcasses and molts, and amorphous aggregates (Mayer, 1993). In the Ross Sea,



aggregates of *P. antarctica*, have been observed to sink at speeds of more than 200 m day$^{-1}$, meaning
they could reach deep water very quickly (Asper and Smith, 1999). In this way, a proportion of the lipid
content of *P. antarctica* and other algae is transported and sequestered in the sediments.

Initial diagenesis is characterized by the preferential degradation of more labile organic compounds e.g.
sugars, proteins, amino acids. Proportionally, lipids are relatively recalcitrant compared to other
biological components and thus are more likely to be preserved as molecular biomarkers on geological
timescales, even where the rest of the organism may be completely degraded. The final proportion of
lipids that are preserved within sediments are affected by factors including the export production, $O_2$
content, residence time in the water column and at the sediment/water interface before deposition,
molecular reactivity, formation of macromolecular complexes, adsorption to mineral surfaces and
bioturbation (Meyers and Ishiwatari, 1993; Killops and Killops, 2004). Within lacustrine sediments, a
significant shift in FA distribution has been shown to occur younger than 100 years due to early
diagenesis, after which the FA distribution remains relatively unaffected by diagenesis (Matsuda, 1978),
thus major changes are assumed to reflect primary environmental signals on longer timescales such as in
our Holocene record. Due to the hyperproductivity of the surface waters offshore Adélie land, we
assume the dominant inputs of the $C_{18}$ FA are from algal sources in overlying waters and upcurrent
regions. Allochthonous inputs e.g. long-range aeolian transport of plant material are assumed to be
minimal.

**4.2.2. Interpretation of hydrogen isotopes**


Compound-specific H isotopes of algal biomarkers are a well-used climate proxy in sediments
throughout the Cenozoic (e.g. Pagani *et al.*, 2006; Feakins *et al.*, 2012). Although diagenetic alteration,
including H-exchange, is possible within sedimentary archives, this has shown to be minimal in
sediments younger than 20 Ma (Sessions *et al.*, 2004). Furthermore, if H-exchange had occurred, we
would expect δ$^2$H values between different FA chain lengths and closely spaced samples to be driven
towards homogeneity, yet large variability remains, suggesting this is not the case. Thus, we are
confident that our measured H isotopes are indicating a primary signal throughout the Holocene.



The $\delta^2$H value preserved in biomarkers is known to be correlated, but offset, with the $\delta^2$H of the water
from which the hydrogen was derived. Measured $\delta^2$H can therefore be described as a function of either
the $\delta^2$H of the water source, or the fractionation occurring between source water and the lipid ($\varepsilon_{l/w}$) (i.e.
vital effects), in which various environmental factors play a part (Sachse *et al.*, 2012).

The main environmental factors controlling $\varepsilon_{l/w}$ are salinity and temperature, with which $\delta^2$H increases
by 1-4‰ per increase in practical salinity unit (psu) (Schouten *et al.*, 2006; Sachse *et al.*, 2012) and
decreases by 2-4‰ per degree C increase (Zhang *et al.*, 2009), respectively. The $\delta^2$H$_{FA}$ record from Site
U1357 displays an absolute range of ca. 123‰, and millennial to centennial scale variability with an
amplitude of ca. 50‰, throughout the core. This would imply extremely large and pervasive variations
in temperature (up to ca. 60°C) and salinity (up to 123 psu) if fractionation driven by either of these
factors were the main control. One study has shown the salinity of present day Adélie shelf waters to
vary between 34 and 34.8 psu (Bindoff *et al.*, 2000), while tetraether-lipid based subsurface (50-200 m)
temperature estimates from nearby Site MD03-2601 (about 50 km west of Site U1357) range from -0.17
to 5.35°C over the Holocene (Kim *et al.*, 2010). Therefore, fractionation changes driven by temperature
or salinity cannot be invoked as a major control on $\delta^2$H$_{FA}$ in the Holocene.

Thus, the most parsimonious explanation relates to changes in $\delta^2$H$_{FA}$ of the water source (Sachse *et al.*,
2012). In the Adélie Basin, the most apparent controls on this are advection, upwelling or inputs of
isotopically depleted glacial meltwater. The $\delta^2$H$_{FA}$ value within Antarctic glaciers is highly depleted
relative to sea water due to the Rayleigh distillation process, leading to highly negative isotope values
for precipitation over the continent.

The glacial meltwater originating from the Ross Ice Shelf is likely to combine ice precipitated
throughout the Holocene and glacial period, and from both the East and West Antarctic Ice Sheets.
However, as noted by Shackleton and Kennett (1975) in their first oxygen isotope record of the
Cenozoic (see their Fig. 6), most of the ice that melts around the margin has been coastally precipitated
(due to higher accumulation rates). Since ice precipitated further inland has a greater residence time
(Shackleton and Kennett, 1975) and significantly lower accumulation rates it will contribute
significantly less to this signal. Thus, the ice that contributed to a marine-based ice sheet collapse along
this margin is best represented by average values of coastal ice dome records at a similar latitude to that





which melted since the LGM (such as TALDICE and Siple Dome) than more southerly locations.
Glacial to Holocene $\delta^2H_{FA}$ values from TALDICE, located on the western edge of the Ross Sea in the
East Antarctic, for example, vary between -276.2 and -330.3‰ (Steig *et al.*, 1998) (converted from $\delta^{18}O$
values following the global meteoric water line (GMWL): $\delta^2H_{FA} = 8.13$ ($\delta^{18}O$) + 10.8), while values
from Siple Dome on the eastern edge of the Ross Sea in the West Antarctic, vary from ca. -200 to -
293‰ (Brook *et al.*, 2005). Taking the average of these values as a rough estimate for the meltwater
gives a $\delta^2H$ value of ca. -275‰. We note our calculations are based on averages of set time periods,
which we expect would integrate ice of various ages - rather than extreme values which could relate to
specific melt events of ice or biases to certain ages/regions. This seems reasonable - the isotopic signal
of coastal surface waters masses advected from the RIS to the Adélie land (as illustrated in Fig. 3 and 4)
must integrate a range of source areas across the RIS and from the coast around to Adélie Land.

In comparison to the highly negative glacial ice isotope composition, sea surface water $\delta^{18}O$
measurements taken near the Mertz Glacier offshore Adélie Land (140-150°E) in summer 2000-2001
ranged between -0.47 and 0.05‰ (Jacobs *et al.*, 2004), equivalent to $\delta^2H$ values of 6.9 to 11.2‰
(average = 9‰) following the GMWL. Thus, the two major hydrogen source pools (RIS glacial ice and
ocean water) have highly contrasting isotope values, meaning inputs of upstream glacial ice could have a
large effect on surface water $\delta^2H$ values in the Adélie Land region.

Taking the average glacial meltwater $\delta^2H$ value as -275‰ and the average modern Adélie surface water
$\delta^2H$ value of 9‰ as end-members, and assuming a biosynthetic offset between the FA and sea water of
173‰ (see below), we can use a simple mixing model to estimate the percentage of glacial meltwater
required in the surface waters to change the $\delta^2H_{FA}$ value to those recorded in U1357B samples. The
most negative values occur during the early Holocene, 11.4 – 8.2 ka, averaging -214.2‰ (n=18) which,
converted to a surface water value of -41‰, requires 17.6% of the surface water to be comprised of
glacial meltwater. During this time, we argue that large volumes of meltwater were reaching the core
site as local glaciers retreated, leading to intense surface-water stratification. Thus, a relatively high
percentage of meltwater in the Adélie Land surface waters seems reasonable. During the mid-Holocene
(5-4 ka), the average $\delta^2H_{FA}$ is very similar (-213.9‰, n=7), requiring 17.2% of the surface water to be
derived from glacial meltwater. During this time, we argue for the dominant meltwater source as coming



from the Ross Sea, and interpret this as a major period of glacial retreat (see section 5.2), during which
large volumes of meltwater are injected into the surface water and transported to the Adélie coast. In
contrast, the most recent samples (last 0.5 ka, n=7), which includes the most positive value of the record,
has an average $\delta^2$H $_{FA}$ value of -174.5‰. This brings the surface water value up to -1.5‰, which
approaches modern measured values, and requires just 3.7% (e.g. well within uncertainties) of the
surface waters in the Adélie Land to be glacial meltwater. However, it is also possible that the meltwater
was dominated by more LGM-aged ice. In either case, perturbation of the exact isotopic values still
indicate only significant changes in the flux of glacial meltwater can account for this signal. For
example, the use of -330‰ (LGM values) for the ice input gives an estimate of 3% of the surface water
being comprised of glacial meltwater for latest Holocene values, and 14.7% for pre 8 ka values. Taking -
240‰ (Holocene values) for the ice input gives an estimate of 4% for latest Holocene values, and 20%
for pre 8 ka values). Thus even with changing isotopic values though the deglacial, this signal of
changing meltwater flux would still dominate. We note these are semi-quantitative estimates, as the
salinity and temperature fractionation could reduce these estimates further (but cannot account for the
whole signal).

Surface water $\delta^{18}$O values around Antarctica (below 60°S), measured between 1964 and 2006, ranged
from -8.52‰ to 0.42‰ (Schmidt *et al.*, 1999), the most negative value having been measured proximal
to the George VI Ice Shelf edge, where high melt rates have been observed (Potter and Paren, 1985). If
converted to $\delta^2$H using the global meteoric water line, these values give a $\delta^2$H range of 83.4‰. Thus,
our absolute $\delta^2$H$_{FA}$ range of 123‰ over the Holocene suggests a range of isotopically depleted
meltwater inputs to our core site over this time that are 1.5 times greater than that occurring in different
locations around the Antarctic in recent decades. This seems plausible seen as geological evidence
indicates large glacial retreat and ice mass loss occurred from the Ross Sea sector during the Holocene
(McKay *et al.*, 2016), meaning resultant changes in surface water are likely to be greater in magnitude
than observed around the Antarctic in recent decades. This assumes a relatively constant value for the
isotopic composition of glacial meltwater, however, there is likely to be some variability due to the
possibility of melting ice of different $\delta^2$H values. But, as discussed above, the meltwater is best
represented by the average values of the ice sheet, rather than extreme values, since it must (over the
broad expanse of the RIS) include an integrated signal, and thus the actual variation in meltwater $\delta^2$H
will be significantly within the range of the end-members.




Although the biosynthetic fractionation of the $C_{18}$ FAs in U1357B is unknown, we assume that the offset
with surface water remains relatively constant throughout the record. Sessions *et al.* (1999) showed the
biosynthetic fractionation of hydrogen isotopes in the $C_{18}$ FA from four different marine algae to range
from -189 to -157‰. If we take the average of these values of 173‰ and apply this as a biosynthetic
offset to the youngest samples in U1357B (last 0.5 ka, n=7), which includes the most positive value of
the record, gives an average $\delta^2 H_{FA}$ value of -174.5‰. This brings the surface water value up to -1.5‰,
which approaches modern measured values (Jacobs *et al.*, 2004).

Furthermore, it is interesting to note that the biosynthetic offsets measured by Sessions *et al.* (1999) for
the $C_{18}$ FA from different algal species have a total $\delta^2 H$ range of 32‰. Although we cannot dismiss
changes in the relative contribution of $C_{18}$ from different species in U1357B samples (and thus different
biosynthetic fractionations), we argue this would only be a minor control on $\delta^2 H$ compared to other
influences. As a thought experiment, taking the above end-members for biosynthetic fractionation from
Sessions *et al.* (1999), even with a 100% change in $C_{18}$ producer to a different algal source, this could
only explain a quarter of the observed $\delta^2 H$ change (i.e. 32‰ of 123‰).

Therefore, we interpret the first order control on $\delta^2 H_{FA}$ at Site U1357 as inputs of isotopically depleted
glacial meltwater. Such inputs are, in turn, influenced by the mass balance of the proximal or up-current
glaciers and ice-shelves.

**5. DISCUSSION**
The stratigraphy of U1357B is divided into three units: the lowermost 10 cm recovered Last Glacial
Maximum (LGM) till (Unit III), overlain by 15 m of laminated mud-rich diatom oozes with ice rafted
debris (IRD) (Unit II), and the upper most 171 m (Unit I) consists of laminated diatom ooze with a
general lack of IRD and a significant reduction in terrigenous sediment (Escutia *et al.*, 2011). The
sedimentology and geometry of the drift prior to ~11.4 ka (Unit II) is consistent with the calving bay
reentrant model (Domack *et al.*, 2006; Leventer *et al.*, 2006) (Fig. 1 and Supplementary Fig. S5;
Supplementary Materials), whereby LGM ice retreated in the deeper troughs while remaining grounded
on shallower banks and ridges. Sediment laden meltwater and IRD content in Unit II (>11.4 ka) is thus





likely derived from local outlet glaciers. However, anomalously old radiocarbon ages due to glacial
reworking precludes development of a reliable age model prior to the Holocene (Supplementary
Materials).

**5.1 Early Holocene**
The base of the drift deposit shows downlapping of material suggesting a supply from the south,
indicating local focusing of meltwater and terrigenous material was the dominating influence until 11.4
ka (Supplementary Materials). This is overlain by onlapping strata (Unit I) with the drift forming an
east-west elongation on the northern flank of the Dumont d'Urville Trough, which is more consistent
with advection of material from the east than with delivery from local outlet glaciers to the south. Thus,
an increased meltwater influence from the Ross Sea is likely since this time (Supplementary Materials).

Due to the potential for competing sources of glacial meltwater in the earliest Holocene, we focus our
study on Unit I, where there is less influence of calving bay processes (Escutia *et al.*, 2011). However,
the earliest part of Unit I (11.4 to 8 ka BP), which includes the most negative $\delta^2 H_{FA}$ values, is
characterized by a very gradual upcore increase of sorting in the terrigenous sediment supply, decreasing
natural gamma ray (NGR) values (Supplementary Fig. S4) and a general lack of IRD. We conservatively
interpret this as potentially maintaining some local glacial meltwater input from local outlet glaciers in
the lowermost interval of Unit I. Nevertheless, this process was probably greatly reduced relative to Unit
II deposition and it is likely much of this signal between 11.4 and 8 ka could still be derived from water
masses advecting to the site from the east (e.g. the Ross Sea).

This is supported by geological and cosmogenic evidence which demonstrates that the Wilkes Land
margin of the East Antarctic, and also the Amundsen Sea margins, had retreated to their modern-day
positions by ~10 ka (Bentley *et al.*, 2014; Mackintosh *et al.*, 2014; Hillenbrand *et al.*, 2017);
Supplementary Materials), thus these margins are unlikely to contribute large scale shifts in meltwater
fluxes to the Adélie Coast during most of the Holocene. Glacial retreat, however, persisted in the Ross
Sea until at least 3 ka (Anderson *et al.*, 2014; Spector *et al.*, 2017) (Supplementary Materials) providing
a large upstream source of meltwater feeding into the Adélie Coast. We therefore interpret our meltwater
signal as being dominated by Ross Sea inputs since at least 8 ka, but potentially as early as 11.4 ka.
Furthermore, the retreat of grounded ice from the outer Ross Sea continental shelf was accompanied by

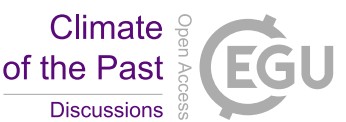

the growth of a significant floating ice shelf (which was not the case in the Amundsen Sea or proximal
East Antarctic coast) (Bentley *et al.*, 2014).

An overall trend to more positive $\delta^2H_{FA}$ values, from the most negative value of the record at ~9.6 ka, to
~8 ka indicates decreasing meltwater (Fig. 2a), thus suggesting a gradually diminished input from either
local outlet glaciers or the Ross Sea. This is associated with an increase in MARs, between 10 and 8 ka,
and is tentatively interpreted to represent the final retreat of residual ice from local bathymetric highs
allowing more material to advect into the drift (Supplementary Fig. S4). Although there is millennial
scale variability, MARs remain relatively high until 4.5 ka. However, $\delta^2H_{FA}$ and MARs show greater
coherence at the millennial-scale after 7 ka BP, suggesting that increased fluxes of glacial meltwater
broadly corresponded to stronger easterly currents, which advected biogenic and terrigenous material
into the drift.

**5.2 Middle Holocene**

A negative excursion in $\delta^2H_{FA}$ starting from 6 ka and culminating at 4.5 ka is interpreted to record a
period of enhanced glacial meltwater flux to the site relating to a final retreat phase of the major ice
sheet grounding line in the Ross Sea embayment (Fig. 5). A marked enrichment of Ba/Ti ratios also
occurs at 4.5 ka, reaching values of 36.1, on a background of baseline fluctuations between 0.1 and 2.7
(Fig. 2g), which suggests enhanced primary productivity, potentially driven by meltwater-induced
stratification. Ongoing Holocene retreat in the Ross Sea is interpreted to be primarily the consequence of
marine ice sheet instability processes resulting from the overdeepened continental shelf in that sector
(McKay *et al.*, 2016).

The $\delta^2H_{FA}$ peak at 4.5 ka in U1357 coincides directly with a rapid shift in HBI biomarker ratios at the
site, as well as sea ice proxies recorded in nearby site MD03-2601, in the Ross embayment (Taylor
Dome ice core on a revised age model) (Steig *et al.*, 1998; Baggenstos *et al.*, 2018) and Prydz Bay
(JPC24) (Denis *et al.*, 2010) (Fig. 2), reflecting a widespread increase in coastal sea-ice concentration
and duration. We interpret decreasing MAR and finer-grained terrigenous content (e.g. increased mud
percent) at Site U1357 after 4.5 ka (Supplementary Fig. S4) to also be a consequence of increased
coastal sea ice, reducing wind stress on the ocean surface and limiting the easterly advection of detritus
to the drift deposit.






Coastal sea-ice concentration and duration remain high throughout the rest of the Holocene (this study
and Steig *et al.*, 1998; Crosta *et al.*, 2008; Denis *et al.*, 2010), compared to the period before 4.5 ka,
despite a decrease in glacial meltwater flux to the U1357 site. In addition, meltwater input prior to 4.5 ka
does not have a major influence on sea ice extent. Thus, an increase in meltwater flux cannot explain the
Neoglacial intensification of sea ice at ~4.5 ka. Here, we propose that greater coastal sea ice cover since
4.5 ka is related to the development of a large ice-shelf cavity in the Ross Sea as the ice sheet retreats
(Fig. 5), which pervasively modified ice shelf-ocean interactions and increased sea ice production.
Models suggest a large cavity on the continental shelf increases contact between basal-ice and
circulating ocean water, driving the formation of a cool, fresh water mass feeding into the AASW,
stabilizing the water column and enhancing the production of sea ice (Hellmer, 2004) (Fig. 5). However,
under small cavities such as in the modern Amundsen Sea influenced by warm-water incursions, ice
shelf melting results in an "ice pump" enhancement of sub-ice shelf circulation. This increases flow of
warm Circumpolar Deep Water (CDW) under the ice shelf that is 100-500 times the rate of melt, and
this volume of water does not allow for supercooling. Small cavity ice shelf outflows are therefore warm
and act to restrict sea ice at the ice shelf front (Jourdain *et al.*, 2017). Thus, during the Holocene, the size
of the cavity must have reached a threshold after which this positive warming feedback switched to a
negative feedback. We argue that such a tipping point takes place at 4.5 ka BP, when our proxy data
suggest meltwater peaks, and would explain why the increase in sea-ice concentration appears rapid and
only occurs at the peak of the meltwater input, and not during its prior increase, or previous meltwater
inputs (Fig. 2a-g).

Although the glacial meltwater volume is greatly reduced after 4.5 ka BP, the volume of Ice Shelf Water
(ISW) produced beneath the modern RIS is estimated at 0.86 Sv-1.6 Sv (Holland *et al.*, 2003; Smethie
and Jacobs, 2005). We note that ISW is not glacial meltwater, but it is defined as a supercooled water
mass formed through interaction with the base of the RIS, but once formed acts to modify other water
masses in the Ross Sea. A significant proportion of ISW is high salinity and is thus advected northwards
at intermediate waters depth to ultimately form AABW. However, a significant volume of ISW is lower
salinity and buoyant, due to development of frazil ice, and acts to mixes with surface waters (Robinson
*et al.*, 2014). Currently, a 0.4 Sv plume of ISW in the western margin of the Ross Ice Shelf (Robinson *et*
*al.*, 2014) is directly delivered to the surface resulting in enhanced sea ice production, while seasonal



melt of this enhanced sea ice further acts to cool and freshen surface waters. Although unrealistic in the
context of a post-LGM meltwater flux from the Ross Sea alone, the larger meltwater release scenarios in
our simulations (0.5 to 1 Sv) show the potential pathways that a cool, fresher surface water mass
collecting and forming on the broad Ross Sea continental shelf would follow (Fig. 3). These waters are
transported in easterly coastal currents to the Weddell Sea and the Antarctic Peninsula. This eventually
retroflects to join the Antarctic Circumpolar Current (Fig. 3b), and thus has potential for cooling and
freshening in the South Atlantic far offshore, as the ice shelf cavity increased in the Ross Sea. Indeed,
offshore site ODP 1094 records increased lithics in the South Atlantic after 4.5 ka (Fig. 2f), relative to
the period before, suggested to have been predominantly transported by sea ice indicating a cooling in
sea surface temperatures and increase in sea-ice extent in the South Atlantic at this time (Nielsen *et al.*,
2007). However, it also is feasible that this circum-Antarctic cooling signal indicates similar melt
processes may have been occurring in the Weddell Sea at ~4.5 ka, as suggested by cosmogenic nuclide
data (Hein *et al.*, 2016).

**5.3 What Drove the Neoglacial Transition?**
Our observed coastal sea-ice increase is part of a widespread transition to Neoglacial conditions both
globally and at high southern latitudes (Kim *et al.*, 2002; Masson-Delmotte *et al.*, 2011; Marcott *et al.*,
2013; Solomina *et al.*, 2015). However, most current climate models do not simulate this cooling trend,
resulting in a significant data-model mismatch (Liu *et al.*, 2014) (Supplementary Fig. S7). Marine ice
sheet retreat along the entire Pacific margin of West Antarctic has previously been proposed to be
triggered by enhanced wind-driven incursions of warm CDW onto the continental shelves in the early
Holocene (Hillenbrand *et al.*, 2017), with continued retreat in the Ross Sea being the consequence of the
overdeepened continental shelf and marine ice sheet instability processes (McKay *et al.*, 2016).
However, we propose that a series of negative feedbacks associated with this retreat and RIS cavity
expansion occurred in the mid-Holocene, with similar processes possibly occurring in the Weddell Sea,
leading to the onset and continuation of Neoglacial conditions. Widespread albedo changes associated
with increased coastal sea ice would have amplified regional cooling trends (Masson-Delmotte *et al.*,
2011), whilst increased stratification resulting from seasonal sea-ice melt and increased ISW production
drove the deepening of the fresher water surface isopycnal at the continental shelf break. Grounding line
retreat creates new space for continental shelf water masses to form, while ice shelf cavity expansion
promotes supercooling and freshening of AASW. Thus, as seasonal sea ice melt and ice shelf





supercooling processes played a greater role in enhancing AASW production on the continental shelf,
they would have acted to restrict warmer subsurface water transport onto the continental shelf (Smith Jr.
*et al.*, 2012) (Fig. 5). Furthermore, the Neoglacial sea-ice increase itself may have been associated with
a stabilising feedback mechanism (which also is not resolved in ice-ocean models) through its role in
dampening ocean-induced wave flexural stresses at ice shelf margins, reducing their vulnerability to
catastrophic collapse (Massom *et al.*, 2018). We suggest that some combination of the above processes
slowed the rate of Ross Sea grounding line retreat (Supplementary Materials) and reduced basal ice shelf
melt as indicated by a trend towards more positive $\delta^2H_{FA}$ values in U1357 between 4.5 and 3 ka (Fig.
2a). Furthermore, large Antarctic ice shelves currently have large zones of marine accreted ice resulting
from supercooling (Rignot *et al.*, 2013), thus the signature of $\delta^2H_{FA}$ is anticipated to become more
positive as the ice shelf approaches a steady state of mass balance, relative to the thinning phases when
basal melt rates exceed that of accretion. The stabilization of $\delta^2H_{FA}$ values observed at 3 ka in U1357
suggests the Ross Ice Shelf has maintained a relatively steady state of mass balance since this time.

A recent study implies an increase in katabatic winds since at least 3.6 ka in the Ross Sea (Mezgec *et al.*,
2017) (Supplementary Materials), leading to enhanced polynya activity. During colder Antarctic
climates, increased latitudinal temperature gradients enhanced katabatic winds in the Ross Sea (Rhodes
*et al.*, 2012). We interpret this katabatic wind and polynya activity signal to be a response to the
preceding Neoglacial cooling at 4.5 ka and evolution of the modern ocean-ice shelf connectivity, which
our data suggest was primarily driven by ice shelf cavity expansion. Furthermore, this transition takes
place on the background of declining winter insolation (Berger and Loutre, 1991) which would have
acted to further enhance and maintain these changes. This insolation decline has previously been
hypothesised as a driver of the Neoglacial increase in coastal sea ice (Denis *et al.*, 2010), however this
monotonic decrease contrasts with the markedly rapid increase in sea ice observed in many records (Fig
2). Our mechanism of ice shelf cavity expansion, reaching a threshold that promoted significant
supercooling of continental shelf water masses, reconciles both the rapidity and timing of Neoglacial
onset in the middle Holocene.

### 6. Conclusions and Implications for Antarctic Climate, Sea-Ice and Ice Shelf Behaviour

The lack of these coupled ice-ocean processes is apparent in recent Earth system model experiments, in
particular the incorporation of evolving ice shelf cavities, with Trace-21k for example, instead





simulating a decrease in Antarctic sea-ice extent and thickness after 5 ka (Supplementary Fig. S7).
These model outputs are in direct contrast to multiple lines of proxy data in this study and previous work
(Steig *et al.*, 1998; Crosta *et al.*, 2008; Denis *et al.*, 2010). Consequently, our results provide insights
into the magnitude of this data-model mismatch, as well as a mechanism for rapid sea-ice change and
grounding line stabilisation on the background of a warming climate (Liu *et al.*, 2014), both on modern
and Holocene time scales. Better representation of the role of evolving ice shelf cavities on oceanic
water mass evolution and sea-ice dynamics, which our data indicate acted as a strong negative feedback,
will be fundamental to understanding the oceanographic and glaciological implications of future ice
shelf loss in the Antarctic.


**Figures**


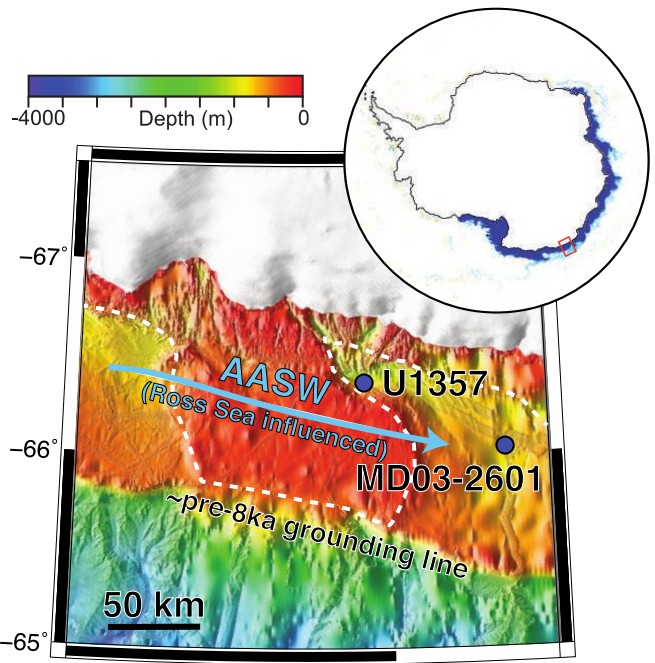


**Figure 1: Location of Sites U1357 and MD03-2601 (blue dots).** The ice sheet grounding line formed a



calving-bay environment (dashed white line) prior to 11.4 ka, but since at least 8.2 ka Antarctic Surface
Water flow is largely advected from the Ross Sea (blue line). Inset map: pathway of freshwater (dark
blue) after 1 year of 1 Sv meltwater released from along the edge of the Ross Ice Shelf in a model
simulation (Supplementary Materials).



**Figure 2: Holocene Adélie Land proxy records from IODP Site U1357 and other circum-Antarctic sites.** Glacial retreat chronologies are shown as bars at the top as discussed in the text. a) $\delta^2$H $C_{18}$ fatty acid at Site U1357 (errors bars based on replicates), with robust locally weighted smoothing (rlowss). b)





*Fragilariopsis curta* group (*F. curta* and *F. cylindrus*) relative abundance at MD03-2601, as a proxy of
sea-ice conditions (Crosta *et al.*, 2008) c) di-unsaturated HBI ($C_{25:2}$; Diene)/tri-unsaturated HBI isomer
($C_{25:3}$; Triene) ratio at Site U1357 d) Methanesulfonate (MSA) concentrations (ppb) from Taylor Dome
ice core e) *F. curta* group relative abundances in core NBP-01-JPC24 f) Coarse lithic (ice-rafted)
content at ODP 1094 (Nielsen *et al.*, 2007) g) Ba/Ti (logarithmic scale) at Site U1357 h) $^{10}$Be
cosmogenic nuclide ages from Scott Glacier in the SW Ross Ice Shelf region (Spector *et al.*, 2017) i)
Temperature signal from principal component analyses of five $\delta^{18}$O records in five East Antarctic ice
cores (Vostok, EPICA Dome C, EPICA Dronning Maud Land, Dome Fuji and Talos Dome) (Masson-
Delmotte *et al.*, 2011).


















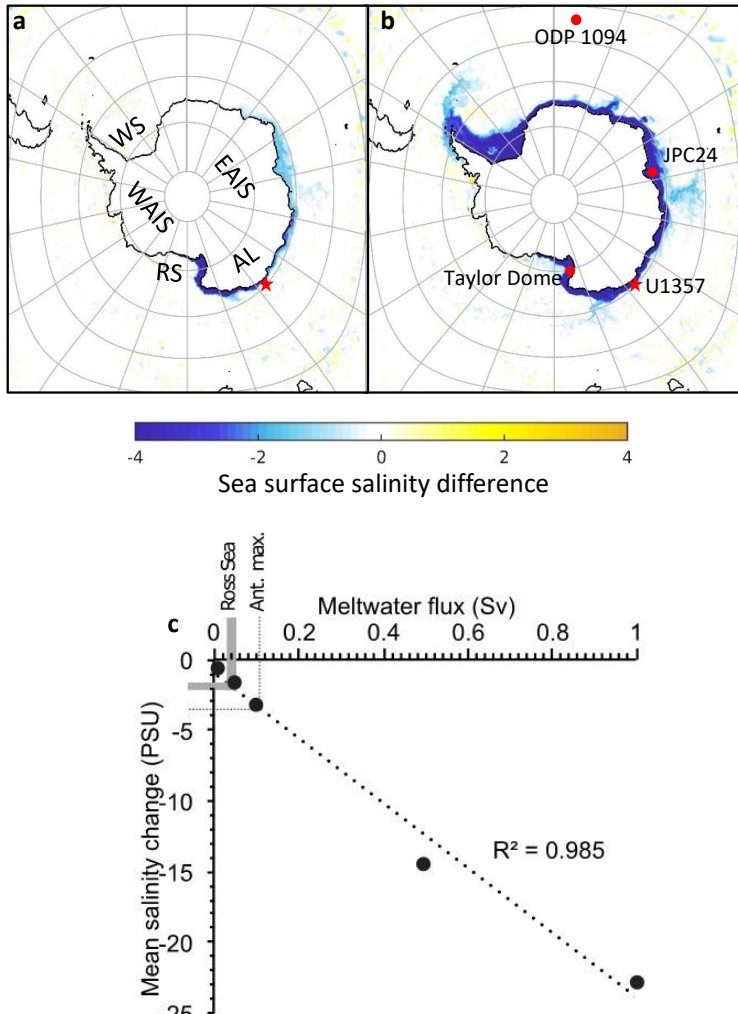

**Figure 3: MITgcm simulations of meltwater release from along the edge of the Ross Ice Shelf**. First
two images show sea-surface salinity difference (in practical salinity units) after 3.5 model years
resulting from meltwater release volumes of a) 0.1 Sv and b) 0.5 Sv. Red star indicates position of Site
U1357 (this study) and red dots show positions of other core sites mentioned in this study where a Mid
Holocene increase in sea ice and/or cooling is recorded: Taylor Dome (Steig *et al.*, 1998; Baggenstos *et
al.*, 2018), JPC24 (Denis *et al.*, 2010) and ODP 1094 (Nielsen *et al.*, 2007). AL = Adélie Land, RS =
Ross Sea, WS = Weddell Sea, EAIS = East Antarctic Ice Sheet, WAIS = West Antarctic Ice Sheet. c)
Scatter plot of simulated meltwater flux (Sv) against mean salinity difference at U1357 core site. Grey





697    band indicates range of plausible Holocene to deglacial Ross Sea meltwater inputs. Dotted line indicates

698    maximum Antarctic meltwater during the Holocene.

699

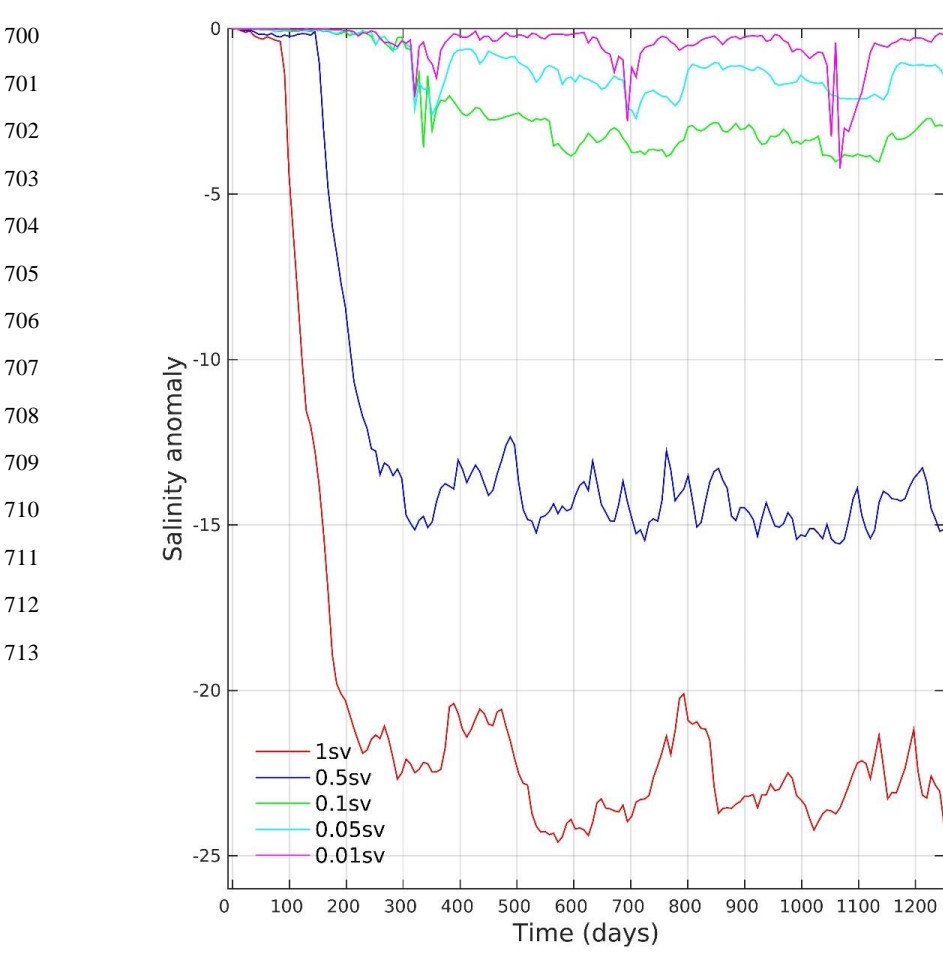

**Figure 4** Simulated salinity anomalies over time at Site U1357 for the five meltwater
release experiments.





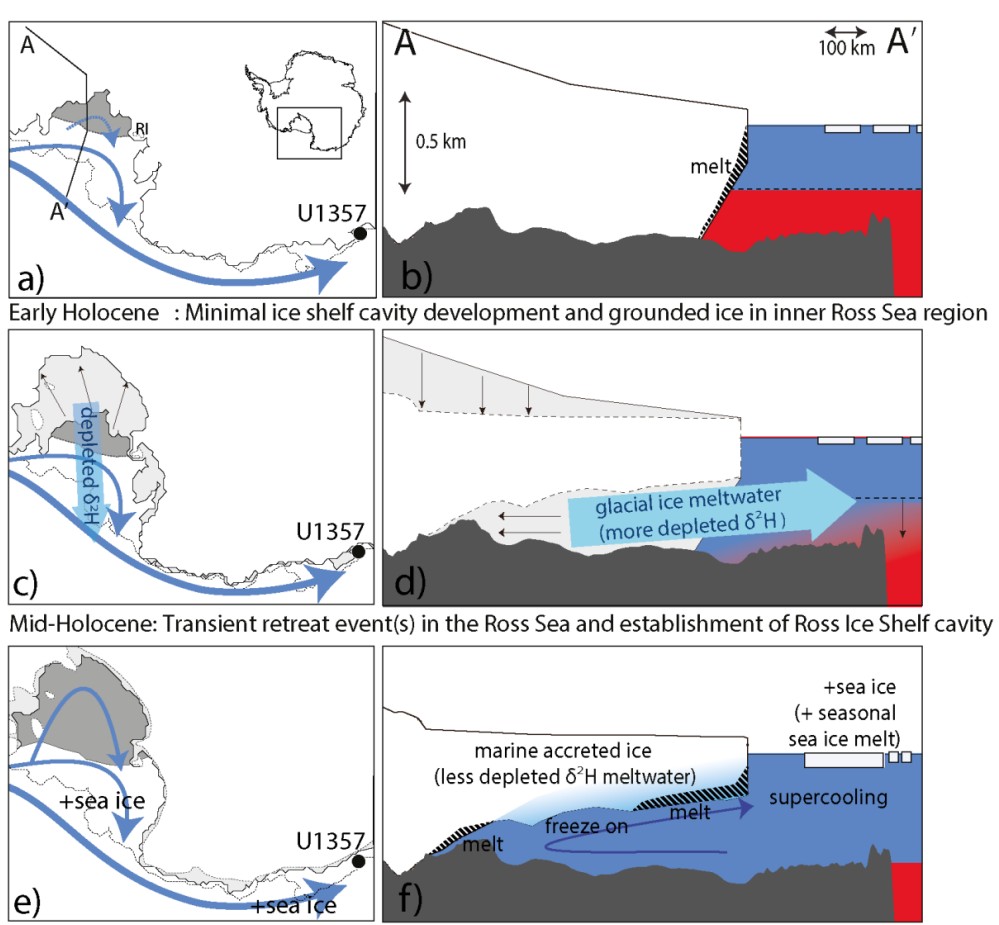

**Figure 5: Conceptual model of evolving Holocene glacial and oceanographic conditions in the Ross Sea region.** Panels on the left show modelled grounding line positions (McKay *et al.*, 2016), and proposed circulation of surface and sub-ice shelf circulating waters (light blue arrows). Panels on the right show cross sections of the Ross Ice Shelf (RIS) and ice-ocean interactions. Dark blue = cool surface waters, Red = warm subsurface waters. a) The grounding line in Adélie Land is near its modern location, but near Ross Island (RI) in the Ross Sea, and ice shelf cavity (dark grey shading) is reduced in size relative to today (McKay *et al.*, 2016). b) Continental shelf profile A-A' (panel a) shows a Ross Sea grounding line in a mid-continental shelf location in close proximity to the RIS calving line (McKay *et al.*, 2016), with subsurface warming on the continental shelf triggering WAIS deglaciation (Hillenbrand *et al.*, 2017). c) Most grounding line retreat south of RI occurred between 9 and 4.5 ka (light grey





shading with black arrows represents area of retreat over this period), proposed to be the consequence of
marine ice sheet instability, but the ice shelf calving line remained near its present position (McKay *et*
*al.*, 2016; Spector *et al.*, 2017). d) Grounding line retreat and ice shelf thinning released meltwater with
negative $\delta^2$H into the surface waters. Increasing ice shelf-oceanic interactions with the development of
the ice shelf cavity (dark grey) led to enhanced Antarctic Surface Water formation; f) Minimal
grounding line retreat has occurred since 4.5 ka, and the RIS supercools AASW leading to enhanced
sea-ice formation despite reduced glacial meltwater flux. Seasonal sea ice meltwater further freshens and
cools AASW. Increased production of AASW on the continental shelf leads to isopycnal deepening
(dotted line) and limits flow onto the continental shelf slowing further grounding line retreat. However,
as the ice shelf is near steady state mass balance and there is a component of marine accreted ice at the
base of the ice shelf (Rignot *et al.*, 2013), the strength of the $\delta^2$H signal is reduced relative to periods of
mass balance loss.

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

**Acknowledgements:** Samples and data were provided by the International
Ocean Discovery Program (IODP). The Natural Environment Research Council funded K.E.A (CENTA
PhD; NE/L002493/1) and J.B. (Standard Grant Ne/I00646X/1). J.B. and O.S. were funded by Japanese
Society for the Promotion of Science (JSPS/FF2/60 No. L-11523). R.M. was funded by the Rutherford
Discovery Fellowship, the NZ Marsden Fund (RDF-13-VUW-003; 15-VUW-131) and Australia-New
Zealand IODP Consortium's Australian Research Council LIEF grant (LE0882854). A.C. was funded
by the NSF (PLR-1443347) and the U.S. Dept. of Energy (DE-SC0016105). A.C. performed model
integrations at the National Research Scientific Computing Center and at XSEDE, an NSF funded



computer center (grant ACI-1548562). C.R. was funded by a L'Oréal-UNESCO New Zealand For
Women in Science Fellowship, University of Otago Research Grant, and the IODP U.S. Science Support
Program. We thank S. Schouten, V. Willmott, F. Sangiorgi, J. Toney and J. Pike for discussions and V.
Willmott, H. Moossen, A. Hallander, R. Jamieson and C. Gallagher for technical support.

**Author contributions:** K.E.A., J.B and R.M. wrote the paper. J.B. and O.S. carried out the fatty acid
isotope analysis, A.A. and R.M. conducted the grain size analyses, J.E. and G.M. generated the HBI
data, F.J.J.E measured X-ray fluorescence scanning and electron microscopy, and C.R conducted the
opal measurements. R.D., R.M., X.C. and G.M. developed the age model. A.C ran the model
simulations. D.P.L and E.G analysed the Trace-21k experiment data. R.D. was lead proponent on the
U1357 drilling proposal. All authors contributed to the interpretations of data and finalization of the
manuscript.

**Competing interests:** The authors have no competing interests.

**Data availability:** There is no restriction on data availability. Upon manuscript acceptance, all
previously unpublished data will be added to the Supplementary Materials and made freely available at
the NOAA NCDC data-base: https://www.ncdc.noaa.gov/data-access/paleoclimatology-data/datasets.


