# Peer review of "FRONT MATTER Title"

_Climate of the Past, 2020_

## Referee Comment (RC1) · Anonymous Referee #1 · 1 Apr 2020

Within their manuscript entitled "Mid-Holocene Antarctic sea-ice increase driven by marine ice sheet retreat", Ashley et al. establish a link between the impact of meltwater discharge from the Ross Sea ice-shelf cavity and sea ice evolution in the Adélie Land coastal region. Based on geochemical, sedimentological and micropaleontological data sets obtained from IODP core U1357 and nearby core MD03-2601, Ashley et al. identify the meltwater signal originating in the Ross Sea and assess its potential role for Holocene sea ice cover at the core site. These proxy reconstructions are complemented by a numerical model simulating the westward circum-Antarctic routing of meltwater released in the Ross Sea sector via the Antarctic Coastal Current. The manuscript is well written and in a very mature state. A major focus is on the application of fatty acid hydrogen isotopes as a proxy for glacial meltwater. Accordingly,

the authors provide an extensive overview on the potential source organisms synthesizing C18 fatty acids and address various aspects relevant for the interpretation of the isotopic values. For background information on and the interpretation of the other proxies applied in this study, the reader is often referred to the supplementary information, which, to my opinion, lowers the readability of the manuscript to a certain extent. Some re-structuring of the manuscript (shifting parts of it into the supplement and vice versa) could help on this. Concerning the paleoenvironmental reconstruction (i.e. the discussion chapter), I miss a more thorough integration and discussion of other already published East Antarctic paleo records (e.g. Berg et al., 2010; Borchers et al., 2016; Kim et al., 2012). While the authors indeed mention Mezgec et al. (2017) at one instance, I miss the actual discussion of their Holocene sea ice reconstructions for the Ross Sea as this could help to link the sea ice evolution in both areas, which would clearly improve the manuscript.

minor points:

lines 56-59: these studies only refer to East Antarctica; re-phrase: "...highlight a major baseline shift in East Antarctic coastal sea ice..."

line 82: diene/triene HBI ratio: please refer to Belt et al. (2016) and provide brief comment that the diene is also called IPSO25 (at least in more recent papers using HBIs for Southern Ocean sea ice reconstructions)

lines 154-170: methods chapter 3.4 (HBIs) should be moved and integrated into chapter 3.1 (Organic geochemical analyses)

line 157: please provide information on the internal standards

lines 261-268: make clear that these simulations are already results of this study

line 327: Tang et al. (2008) do not state that P. antarctica exists within sea ice; please provide an appropriate reference

line 342: please provide reference for better preservation of biomarker lipids compared

to microfossil remains

lines 393-394: unclear - what is meant with "...ice that contributed to a marine-based ice sheet collapse along this margin..."

lines 492-529: what do the HBIs reflect in terms of sea ice cover during the Early Holocene?

References

Berg, S., Wagner, B., Cremer, H., Leng, M. J., and Melles, M., 2010, Late Quaternary environmental and climate history of Rauer Group, East Antarctica: Palaeogeography, Palaeoclimatology, Palaeoecology, v. 297, no. 1, p. 201-213.

Borchers, A., Dietze, E., Kuhn, G., Esper, O., Voigt, I., Hartmann, K., and Diekmann, B., 2016, Holocene ice dynamics and bottom-water formation associated with Cape Darnley polynya activity recorded in Burton Basin, East Antarctica: Marine Geophysical Research, v. 37, no. 1, p. 49-70.

Kim, J.-H., Crosta, X., Willmott, V., Renssen, H., Bonnin, J., Helmke, P., Schouten, S., and Sinninghe Damsté, J. S., 2012, Holocene subsurface temperature variability in the eastern Antarctic continental margin: Geophysical Research Letters, v. 39, no. 6.

---

## Referee Comment (RC2) · Anonymous Referee #2 · 4 Apr 2020

Review of Ashley et al. This study presents a new Holocene record (12 ka to present) from the Adelie basin to infer past changes in meltwater from Antarctica as well as sea-ice changes. Hydrogen isotopes of fatty acids, relative abundance of phytoplanktons, and organic compound composition are measured in the marine sediment core. Other measurements were made (e.g. grain size analysis), but are not shown in the main text. In addition, meltwater experiments are performed with an eddy-permitting ocean model. The study is interesting, presenting a lot of information, which can help in the understanding of the deglaciation of Antarctica. I think in between the main text and Supp. all the information is there. However, the manuscript needs to be significantly restructured as it is currently very hard to follow. The reasoning and result from each analysis/modelling needs to be more clearly laid out. This is developed in the

comments below:

1) Numerical modelling: I was quite excited at first to see a series of simulations performed with 1/6deg model. However, the results are only very briefly described in the paragraph L. 261. The motivation behind the modelling experiments should be more clearly explained as well as the limitations/assumptions taken. The volume of ice equivalent to each meltwater input should be given. They have the advantage of being performed with a high-resolution ocean model, however for the problem at hand (understanding the percentage of meltwater coming from the Ross Sea, without the use of water isotopes), they are a bit limited. The paragraph L. 261 surprised me, as the setting of the study is described, and suddenly some results of the numerical simulations are described. Until looking at the figures, it was very unclear to me that you were referring to your own simulations. Please be more specific, or consider restructuring, also because Figures 3 and 4 are (very briefly) described, whereas Figure 2 has not been called yet.

2) Based on the Methods, a lot of analyses have been made on the sediment core, but i) only d2H of fatty acids, phytoplankton %, and organic compound composition are shown in the main text, ii) only hydrogen isotopes of fatty acids are presented in the "results"(there is in fact no "result" section), iii) most of the other analyses are in fact presented in the "Discussion" section and the supplementary. As reading L. 536 in the Discussion, I realized you were in fact talking about your results (Ba/Ti). Searching through the document I realized this was briefly mentioned elsewhere, but this should be made much more obvious. Methods could be shorter but by going to the point of each measurement that you need for your interpretation. It would be good to have all the necessary/needed sediment analyses and modelling results presented in the "Results" section as well as in the Figures of the main text. The discussion section should then focus on the bigger picture and putting the results within the context of previous studies.

Suddenly in the Discussion section (e.g. L. 544, L. 550), conclusions are presented

about changes in sea-ice, without knowing where this is coming from. Please clearly state in the results section how you infer the changes in sea-ice and what the main changes across the Holocene are.

It is not clear to me that all the other proxies (from other marine sediment/ice cores) presented in Figure 2 are consistently discussed in the text. Please make sure to clearly mention what each proxy suggest/represent (i.e. how to interpret changes in MSA, Lithics… might not be straight forward for all readers), and refer to it as Fig. 2f, 2g, with the appropriate reference.

3) The manuscript is quite well referenced, particularly with respect to the setting of the study, but I am surprised (particularly given the co-author list) not to see any comparison or discussion with previous modelling work on Antarctic deglaciation. Even though these simulations (e.g. Golledge et al., 2014) are associated with significant uncertainties, they might help in discussing the origin and magnitude of Antarctic meltwater.

4) Minor points and typos: L.101: "sealed"

L. 191: Please use present tense.

L. 257: Please correct the typo "10ˆ6" and add Sv, so it should read (1 Sv=10ˆ6 m3/s).

L. 257: Maybe add a caveat to the "76 Sv", which seems a bit high. In Thompson et al., 2018 (Review of Geophysics on the ASC), they state that Pena-Molino et al., (2016) find a highly variable ASC at 113E from 0 to 100 Sv, but with a mean of 21 Sv.

L. 259: "the gyre transport is around"

L. 268: "of the meltwater input."

L. 347: there is something wrong with that sentence.

L. 597-598: "most models", There are not many transient simulations of the Holocene currently published, and the one you refer to could be the only one. So instead of "most models", simply state the "TRACE21 simulation"

---

## Author Comment (AC1) · 21 May 2020

**Ashley et al. response to Anonymous Referee #1 (our replies are in bold)**

Within their manuscript entitled "Mid-Holocene Antarctic sea-ice increase driven by marine ice sheet retreat", Ashley et al. establish a link between the impact of meltwater discharge from the Ross Sea ice-shelf cavity and sea ice evolution in the Adélie Land coastal region. Based on geochemical, sedimentological and micropaleontological data sets obtained from IODP core U1357 and nearby core MD03-2601, Ashley et al. identify the meltwater signal originating in the Ross Sea and assess its potential role for Holocene sea ice cover at the core site. These proxy reconstructions are complemented by a numerical model simulating the westward circum-Antarctic routing of meltwater released in the Ross Sea sector via the Antarctic Coastal Current. The manuscript is well written and in a very mature state. A major focus is on the application of fatty acid hydrogen isotopes as a proxy for glacial meltwater. Accordingly, the authors provide an extensive overview on the potential source organisms synthesizing C18 fatty acids and address various aspects relevant for the interpretation of the isotopic values.

**We thank the reviewer for this positive and constructive comment.**

For background information on and the interpretation of the other proxies applied in this study, the reader is often referred to the supplementary information, which, to my opinion, lowers the readability of the manuscript to a certain extent. Some re-structuring of the manuscript (shifting parts of it into the supplement and vice versa) could help on this.

**We thank the reviewer for pointing this out and agree that due to the large amount of information and data included in the paper, much information was moved into the supplementary information which may have become confusing at times. In the final submission we will move some more information into the main manuscript, including a new 'results' section which describes the results from our main datasets to make it easier for the reader to understand what new data we are presenting without having to look to the supplementary. In addition, we recognise that the reader is often referred to Fig S2, which contains our grainsize data. Therefore, we will move this from the supplementary into the main text, so that all our original datasets are displayed in the main manuscript to make it easier for the reader.**

Concerning the paleoenvironmental reconstruction (i.e. the discussion chapter), I miss a more thorough integration and discussion of other already published East Antarctic paleo records (e.g. Berg et al., 2010; Borchers et al., 2016; Kim et al., 2012). While the authors indeed mention Mezgec et al. (2017) at one instance, I miss the actual discussion of their Holocene sea ice reconstructions for the Ross Sea as this could help to link the sea ice evolution in both areas, which would clearly improve the manuscript.

**We will be happy to include a wider discussion of East Antarctic paleoenvironment changes in our final submission including the papers the reviewer has mentioned. However, we note that the study by Berg et al. (2010) was performed in the Rauer Group, an archipelago very close at the Prydz Bay coast, where condition are strongly influenced by local processes, linked to the presence of the islands and nearby coast. It is therefore perhaps less likely to be representative of the wider environment than other core sites.**

minor points:
lines 56-59: these studies only refer to East Antarctica; re-phrase: "...highlight a major baseline shift in East Antarctic coastal sea ice..."
**We will amend this in the final revised submission.**

line 82: diene/triene HBI ratio: please refer to Belt et al. (2016) and provide brief comment that the diene is also called IPSO25 (at least in more recent papers using HBIs for Southern Ocean sea ice reconstructions)
**We will add reference to this in the final revised submission.**

lines 154-170: methods chapter 3.4 (HBIs) should be moved and integrated into chapter 3.1 (Organic geochemical analyses)
**We will move this section in the final revised submission**
line 157: please provide information on the internal standards
**We will add this information this in the final revised submission**
lines 261-268: make clear that these simulations are already results of this study
**We will amend this in the final revised submission by explaining the reason for the model simulations before presenting the results.**
line 327: Tang et al. (2008) do not state that P. antarctica exists within sea ice; please provide an appropriate reference
**We will add a more appropriate reference for this in the final revised submission**
line 342: please provide reference for better preservation of biomarker lipids compared to microfossil remains
**We will add an appropriate reference for this in the final revised submission**
lines 393-394: unclear - what is meant with "...ice that contributed to a marine-based ice sheet collapse along this margin..."
**This is referring to the ice that was melting along this margin. We will rephrase this in the final revised submission to make it clearer.**
lines 492-529: what do the HBIs reflect in terms of sea ice cover during the Early Holocene?
**The HBI diene/triene ratio is very low during this period due to very low concentrations (or absence) of the HBI diene. While this does suggest that sea ice cover was low in comparison to the Late Holocene, the low concentrations mean we cannot confidently interpret any trends during this period.**

---

## Author Comment (AC2) · 21 May 2020

**Ashley et al. response to Anonymous Referee #2 (our replies are in bold)**

Review of Ashley et al. This study presents a new Holocene record (12 ka to present) from the Adelie basin to infer past changes in meltwater from Antarctica as well as sea-ice changes. Hydrogen isotopes of fatty acids, relative abundance of phytoplanktons, and organic compound composition are measured in the marine sediment core. Other measurements were made (e.g. grain size analysis), but are not shown in the main text. In addition, meltwater experiments are performed with an eddy-permitting ocean model. The study is interesting, presenting a lot of information, which can help in the understanding of the deglaciation of Antarctica. I think in between the main text and Supp. all the information is there. However, the manuscript needs to be significantly restructured as it is currently very hard to follow. The reasoning and result from each analysis/modelling needs to be more clearly laid out.

**We thank the reviewer for this constructive review. We will restructure the manuscript in the final submission by moving some information from the supplementary into the main text to make it easier for the reader to follow.**

This is developed in the comments below:

1) Numerical modelling: I was quite excited at first to see a series of simulations performed with 1/6deg model. However, the results are only very briefly described in the paragraph L. 261. The motivation behind the modelling experiments should be more clearly explained as well as the limitations/assumptions taken. The volume of ice equivalent to each meltwater input should be given. They have the advantage of being performed with a high-resolution ocean model, however for the problem at hand (understanding the percentage of meltwater coming from the Ross Sea, without the use of water isotopes), they are a bit limited. The paragraph L. 261 surprised me, as the setting of the study is described, and suddenly some results of the numerical simulations are described. Until looking at the figures, it was very unclear to me that you were referring to your own simulations. Please be more specific, or consider restructuring, also because Figures 3 and 4 are (very briefly) described, whereas Figure 2 has not been called yet.

**We thank the reviewer for prompting us to improve the organisation of the modelling elements. In our final submission we will describe and explain the reasoning for the modelling study more clearly. As the reviewer points out, the model results are briefly mentioned in L. 261, but we will remove this reference as it is confusing here. We will include a new 'Results' section in the final submission, following Section 4, in which the model results are explained and described in detail. Here we will make it clear that they are our own simulations and explain our reasoning behind them. The simulations will then be described in the context of other data in the Discussion section. In addition, we will swap Fig. 2 and 3 around so that the model results are presented before the other datasets following the order they are described in the main text. The reviewer suggests that the modelling is a bit limited as it does not include water isotopes. However, the main purpose of the simulations is to help understand the pathway of glacial meltwater released from the Ross Ice Shelf and interactions with the Adélie Land coast. We would say that the model simulations are sufficient for this purpose.**

The volume of ice equivalent to each meltwater input should be given.

**The volume of ice equivalent to meltwater inputs is illustrated, in broad terms, by the grey bars in Fig. 3. E.g. A meltwater flux of ca. 0.1 (Sv) is equivalent to a major circum-Antarctic melting event, as suggested by Golledge et al., (2014; Nat. Comms). Most of this meltwater would have originated from the Ross and Weddell Sea during the Holocene. Higher fluxes (e.g. 0.5 and 1 Sv) are relevant in the context of the supercooled ISW water and freshening of surface waters that occur on the**

broad continental shelf the Ross Ice Shelf, that was previously much narrower. This is not meltwater per se, but supercooled waters emanating from cavities can act like meltwater, in terms of buoyancy and pathways. For example, Robinson et al. (2014; JGR) model 0.4 Sv ISW from the McMurdo Ice Shelf which directly passes into surface waters and directly influencing sea ice growth. Determining the exact flux of Ross Sea ice shelf *influenced* surface water proves difficult due to mixing/modification process and sparse observational data.  In our revised manuscript we will expand the description in the figure legend and will more clearly cross reference to the main text of the manuscript to clarify these meltwater scenarios.

2) Based on the Methods, a lot of analyses have been made on the sediment core, but i) only d2H of fatty acids, phytoplankton %, and organic compound composition are shown in the main text, ii) only hydrogen isotopes of fatty acids are presented in the "results"(there is in fact no "result" section), iii) most of the other analyses are in fact presented in the "Discussion" section and the supplementary. As reading L. 536 in the Discussion, I realized you were in fact talking about your results (Ba/Ti). Searching through the document I realized this was briefly mentioned elsewhere, but this should be made much more obvious. Methods could be shorter but by going to the point of each measurement that you need for your interpretation. It would be good to have all the necessary/needed sediment analyses and modelling results presented in the "Results" section as well as in the Figures of the main text. The discussion section should then focus on the bigger picture and putting the results within the context of previous studies.

**We are not sure what the reviewer means** 'only d2H of fatty acids, phytoplankton %, and organic compound composition are shown in the main text'. **In the introduction (L. 69 – 86) we introduce the new measurements made on U1357, specifically:** *"Here, we present a new Holocene record of glacial meltwater, sedimentary input and local sea ice concentrations from Site U1357 using compound-specific hydrogen isotopes of fatty acid biomarkers (d2HFA), terrigenous grain size, biogenic silica accumulation, highly-branched isoprenoid alkenes (HBIs) and Ba/Ti ratios (Fig. 2 and S4)."* **These new data are shown in the main data Figure (Fig. 2) along with data-sets from the literature. Never-the-less, in our final submission we will include a clearer Results section, following on from Section 4, in which all the new data presented in this paper will be described in more detail. This will help the reader understand which datasets are new. In addition, we will move Fig S2 (displaying sedimentary data from the core) from the supplementary into the main manuscript so that all of our data is displayed in the main text. To make it clear which data we our presenting in the paper, we will also add some additional text to the end of Section 4 explaining the interpretation of our other proxy data, in addition to the hydrogen isotopes.**

**In our final submission, we will also try to cut down the methods section. However, as we are presenting a lot of new data, there are a lot of methods to describe and we do not think there is a lot of detail which can be cut out while still keeping enough information so that the reader can see that our methods are valid and reproducible.**

Suddenly in the Discussion section (e.g. L. 544, L. 550), conclusions are presented about changes in sea-ice, without knowing where this is coming from. Please clearly state in the results section how you infer the changes in sea-ice and what the main changes across the Holocene are.

**We agree that our interpretation of the HBI proxy could be made clearer and in our final submission we will include this interpretation at the end of Section 4. We will also include a brief description of the HBI data in the Results section.**

It is not clear to me that all the other proxies (from other marine sediment/ice cores) presented in Figure 2 are consistently discussed in the text. Please make sure to clearly mention what each proxy suggest/represent (i.e. how to interpret changes in MSA, Lithics… might not be straight forward for all readers), and refer to it as Fig. 2f, 2g, with the appropriate reference.

**We will include specific reference to each dataset in the main text and refer to the relevant part of the figure where it is displayed.**

3) The manuscript is quite well referenced, particularly with respect to the setting of the study, but I am surprised (particularly given the co-author list) not to see any comparison or discussion with previous modelling work on Antarctic deglaciation. Even though these simulations (e.g. Golledge et al., 2014) are associated with significant uncertainties, they might help in discussing the origin and magnitude of Antarctic meltwater.

**We refer the reviewer to our Response 1, as the grey bars in Fig. 3 are based on previous modelling work (Golledge et al., 2014). We apologise that this was not properly discussed, in our revised manuscript we will expand this discussion and will include relevant references (e.g. Golledge et al., 2014; Robinson et al., 2014).**

4) Minor points and typos:
L.101: "sealed"
L. 191: Please use present tense.
L. 257: Please correct the typo "10ˆ6" and add Sv, so it should read (1 Sv=10ˆ6 m3/s).
**We will amend the above typos in the final revised submission.**

L. 257: Maybe add a caveat to the "76 Sv", which seems a bit high. In Thompson et al., 2018 (Review of Geophysics on the ASC), they state that Pena-Molino et al., (2016) find a highly variable ASC at 113E from 0 to 100 Sv, but with a mean of 21 Sv.

**Agreed, we will add the caveat as suggested.**

L. 259: "the gyre transport is around"
L. 268: "of the meltwater input."
L. 347: there is something wrong with that sentence.
**Agreed. The sentence should read "A significant shift in FA distributions has been shown to occur within 100 years due to early diagenesis". This will be corrected in the final submission.**
L. 597-598: "most models", There are not many transient simulations of the Holocene currently published, and the one you refer to could be the only one. So instead of "most models", simply state the "TRACE21 simulation"
**We will amend this in the final revised submission.**

---

## Author Response (AR2)

**Response to editor comments (our replies in bold)**

Dear Kate E. Ashley et al.,

Thank you for submitting your revised version of your manuscript "Mid-Holocene Antarctic sea-ice increase driven by marine ice sheet retreat". Overall, I think you have done a very good job addressing the reviewer's comments. Before accepting the paper, I have a few minor comments that I would like you to address.

L56: In your responses to the reviewer 1 you stated that you would change from Antarctica to East Antarctica. I cannot see that this have been done.

**We have corrected this (line 55).**

L600: Please specify to where in the supplementary material you refer the reader.

**We had added reference to Supplementary section S2.2 and Fig. S4 (line 535).**

L609: Here you refer to IRD, but with no reference to any of the figures or to literature. Please add appropriate reference. In you results section you mention IRD, but purely with a reference to a published paper and with no reference to any figures where the data is shown. So, is IRD part of your results or not?

**In lines 489-491 we state that there is a lack of IRD in unit 1 (Escutia et al., 2011), which is the focus of this paper. Thus, there is no data to show but we have added in a reference to Escutia et al., 2011 (lines 544-545).**

L688: Instead of referring to "this study ", refer to the figure showing the relevant data. Specify what type of proxies the other studies you use to support this statement is based on.

**We have removed 'this study' and refer specifically to our HBI data (Fig. 4c); sea ice diatoms in core MD03-2601 (Crosta et al., 2008);  methanesulfonic acid concentration in Taylor dome ice core (Steig et al., 1998); and sea ice diatoms in core JPC24 (Denis et al., 2010. See lines 621-624.**

L765: Delete the first "patterns"?

**We have edited this as suggested (line 704)**

L735: In your response to reviewer 2 you stated that you would be more specific and instead of "most models" refer to "TRACE21". This has not been done. Furthermore, why not include Figure S6 in the manuscript instead of as a supplement as long as it is discussed here?

**It is true that we originally suggested we would change that, however, we realised that as well as TRACE21, we also refer to the study by Lui et al., 2013 which analyses three coupled ocean–atmosphere models (CCSM3, FAMOUS and LOVECLIM), thus we are not only referring to the TraCE-21k simulation. We have now included Fig S6 in the main manuscript (Fig. 7)**

Many of your figures are build-up of several panels/sub-figures (a, b, c,…). Occasionally you refer to the individual panels, but mostly you refer to the full figures without further specifications. Please check that whenever you only refer to one or a few of the datasets in a figure, specific with panels you refer to.

**We have been through the manuscript and referred to the individual panels within figures where required.**

Best regards,
Bjørg Risebrobakken
Editor Climate of the Past

[revised manuscript text omitted]

**SUPPLEMENTARY MATERIALS**

**S1. Age model**

We developed an age model for core U1357B based on 87 $^{14}$C analyses on bulk organic carbon (Fig. S1). In the standard IODP CSF-A depth scale, recovery often exceeds 100% and to correct for this, the standard IODP procedure is to apply a linear compression algorithm which is based on the assumption that expansion is uniform in the core. However, in U1357B, expansion due to biogenic gas was particularly high and resulted in discrete sections of core being pushed apart creating voids in the depth scale that did not represent real gaps in the stratigraphy. To account for this, the voids are numerically removed and depth scale adjusted prior to linear compression being applied (if recovery still exceeds 100%).

The model was calibrated with a reservoir age correction of 1200±100 years and depth to age conversions achieved by using BACON. This is a Bayesian iteration scheme that invokes memory from dates above any given horizon, and produces a weighted mean and median age-depth curve

[Figure]

**Figure S1**. Age-depth plot of U1357B, using the default outputs from the BACON software. Upper panels show (from left to right): a stationary distribution of the Markov Chain Monte Carlo iterations; prior (green curve) and posterior (grey curve) distribution for the accumulation rate; prior (green curve) and posterior (grey curve) distribution for memory. Bottom panel shows the calibrated [14]C dates (blue) and age depth model with 95% confidence intervals.

(Blaauw and Christeny, 2011). The top depth of 3 m is consistent with the reservoir age in the Southern Ocean (Hall et al., 2010). Bulk organic carbon ages in the Antarctic are commonly compromised by reworking of older carbon in the sediment column (Andrews et al., 1999), which is compounded by extreme sediment starvation of post-LGM sequences in the Antarctic. However, due to extremely high input of autochthonous carbon associated with the Adélie Drift deposit, which is a predominately seasonally deposited

[Figure]

**Figure S2**. Age-depth plot of MD03-2601, using the default outputs from the BACON software. Upper panels show (from left to right): a stationary distribution of the Markov Chain Monte Carlo iterations; prior (green curve) and posterior (grey curve) distribution for the accumulation rate; prior (green curve) and posterior (grey curve) distribution for memory. Bottom panel shows the calibrated [14]C dates (blue) and age depth model with 95% confidence intervals.

diatom bloom, this reworking is expected to be minimal at the U1357 site, and is only a potential issue at the base of the core (Unit II), due to increased proximity to glacial influences from the

Adélie Land coast during the deglaciation (e.g. > 11.4 ka). This is supported by the radiocarbon ages, maintaining a strong stratigraphic order (within error), relatively consistent sedimentation rates throughout the deposited interval, and core top ages that are consistent with the expected reservoir age.

We also recalibrated the age-model for MD03-2601 applying the BACON methodology using the [14]C dates presented in (Crosta et al., 2008) (Fig. S2). This age model differs from that of (Denis et al., 2009a), who discarded two [14]C ages bracketing the mid-Holocene (4.4 and 5.6 cal ka BP), on the basis of an inferred meteorite impact at ca. 15 m and correlated this to an event at 4 ka BP. We note that this impact correlation does not provide a unique absolute age constraint, with our revised age model instead indicating an age of 5.4 cal ka BP for this impact event. Critically, comparison between the age model for U1357B and the revised age model for MD03-2601 now indicates a strong similarity regarding changes in the sedimentation rates at ca. 4.5 ka and ca. 2 ka BP, indicating that both sites are influenced by similar depositional processes.

**S2. Further details on interpretation of proxy data**

**S2.1    Ba/Ti ratio excess as a primary productivity proxy**

Ba-based proxies (e.g., Ba/Ti or equivalent Ba/Al) in the Wilkes Land margin sediments have been commonly related to marine productivity (Presti et al., 2011), although studies in other pelagic environments indicate that they can also be sensitive to bottom current intensity (Bahr et al., 2014), meltwater (Plewa et al., 2006), and other processes (Griffith and Paytan, 2012). FESEM analysis and images at Site U1357 indicate the presence of biogenic barite (Fig. S3a), recognized by the elliptical morphologies and sizes between 1 to 3 μm (Paytan et al., 2002). Titanium is found associated with small heavy minerals (ilmenite) with angular and low sphericity shapes.

Along the entire record, Ba/Ti ratios show persistent periodic fluctuations with values between 0.1 and 2.7. Nevertheless, a marked enrichment can be observed at 4.5 ka reaching Ba/Ti ratio values over 36.1 (Fig. 2). Pore water analysis indicates that the carbon dioxide (methanic) reduction zone (CRZ) is reached just few cm near sea floor and the upper 20 m already contain sulfate-free interstitial waters (Escutia et al., 2011). Observed geochemical conditions indicate that some Barite dissolution could be expected, but there is no diagenetic barite that could justify the obtained enrichment. In addition, at the enriched interval we did not observe any lithological change or enrichment in other elements (e.g., Si). The influence of aeolian dust or fluvial input on Ba input can also be discarded in the glaciated Wilkes Land margin. In the same way, Ba concentration in sea-ice is considered null because on an annual cycle, sea ice does not constitute a net source or sink of these species to the underlying seawater (Thomas, 2011).

Furthermore, FESEM imaging detected biogenic barite during intervals were the Ba/Ti excess occurs (Fig. S3) pointing to an increase in productivity. This may be driven by water column stratification

[Figure]

**Figure S3.** Authigenic marine barite (red circle, size > 300nm) observed in Ba/Ti enriched interval (sample U1357B-8H-2A 141-143; age: 4,430 cal yr) a) SEM image obtained with secondary electrons with Inlens detector at 20 kV b) SEM-EDX spectrum (analyzed spot marked with a white cross in image c)) showing barite composition ($BaSO_4$) c) Same barite shown in a backscattered electron (BSE) mode by AsB detector at 20 kV.

or greater nutrient availability. This interpretation is coherent with other paleoproductivity reconstructions in this area, in particularly peaks in [230]Th-normalized fluxes of biogenic silica (BSi) and organic carbon content recorded in nearby core MD03-2601 (Denis et al., 2009b) (when using the recalibrated age-model – see S2).

**S2.2 Grain size**

Grainsize analysis was conducted on paired samples with lipid biomarker samples in Unit I. Unit I represents the onset of the modern deposition at Site U1357, and the underlying stratigraphy is discussed in the main text. Between ca. 11.4 and 8 ka, U1357B has a relatively high terrigenous component (i.e. high Natural Gamma Radiation (NGR) content and low BSi%; Fig 4). The grain size distribution contains coarse tails of fine (125-250 μm) to medium sands (250-500 μm), but only one sample contains coarse sands (>500 μm) that may represent ice-berg rafted debris (IBRD). However, terrigenous content and IBRD is more common in the underlying Unit II, which is interpreted to represent a true "calving bay environment" (Escutia et al., 2011). Shipboard description of the core faces found the presence of small isolated facetted and striated pebbles (lonestones) in Unit II (Escutia et al., 2011), which is supportive of an iceberg component to sediment supply, but these are largely absent in Unit I. The fine grained sands and muds have a distribution with similar modes to overlying intervals, albeit with an increase in the size of the coarse silt and very fine sand modes. The subtle increased sorting up core between ca. 11.4 and ca.8 ka (from very poorly to poorly sorted, Fig. 4) is consistent with an increasingly more distal setting, with less potential for a glacial grounding line sediment supply (Powell and Domack, 1995; McKay et al., 2009). This interval is interpreted to reflect the final post-LGM retreat of local EAIS outlet glacier grounding lines from a proximal (less sorted, more IBRD) to more distal setting (better sorted, less IBRD) from the site, although we note a much larger shift occurs at the Unit I/II boundary at 170.25 mbsf (ca.11.4 ka) (Escutia et al., 2011) and dominant sediment supply from local outlet glaciers probably ceased at this earlier time. It is likely this distal setting was close to the modern day grounding line, as the Dumont d'Urville Trough is overdeepened between Site U1357 and the modern day grounding line, and this bathymetry configuration is inherently unstable for marine-based ice sheets (Thomas and Bentley, 1978; Bentley et al., 2014).

Today, Adélie Land glaciers are inferred to have relatively clean basal layers due to the solid bedrock (Kleinschmidt and Talarico, 2000), and distal polar glacimarine settings are usually sediment starved and provide very low inputs of terrigenous sediments (Powell and Domack, 1995; McKay et al., 2009).

There are also no large proglacial fans evident at the mouths of these glaciers (Beaman et al., 2011) and consequently, direct sediment discharge from the Mertz and Ninnis Glaciers is unlikely to be of significant quantity to sustain dense overflows delivering sediment over the Adélie Bank and into the Dumont d'Urville Trough. Therefore, the release of terrigenous material through glacial melting is low (to absent) when glacier activity is steady and distal from the site, but is anticipated to increase with increased proximity to the glacier front or with enhanced dynamic ice discharge (which may occur either during a retreat or advance), which would be associated with an increase in IBRD from local outlet glaciers. We note evidence for proglacial fan deposition and IBRD is lacking throughout Unit I (post 11.4 ka).

Between 9 and 4.5 ka, mass accumulation rates (MARs) (both biogenic and terrigenous; Fig. S4) are relatively high, albeit with millennial scale variability. However, the mean grain size and sorting of the terrigenous material is relatively stable throughout the entire interval, and as with the rest of Unit I there is an almost complete lack of IBRD. This suggests that sediment input from grounding line processes were also minimal through this time. Based on the drift morphology (Fig S4), and the prevailing easterly flow of the Antarctic Coastal Current, this interval is interpreted to be the result of sediment advection to the site from the east as residual ice retreated from the bathymetric highs in the region. Diatom frustules and sponge spicules are mainly in the 16 to 63 μm range, much of which is maintained in suspension by weak currents (a few cms$^{-1}$) (Dunbar et al., 1985). The greater area of open water for primary production during the summer, combined with an open pathway for advection of biogenic matter from the MGP (Fig. 1) can thus explain the significant rise in linear sedimentation and mass accumulation rates for both biogenic and terrigenous material. Most of the terrigenous material after ~8 ka is proposed to have been primarily eroded off the Adélie Bank by westward flowing currents into the Adélie drift where sediment would have settled out from suspension. Terrigenous sediment younger than 8 ka in this drift is almost entirely finer than 125 µm, while Dunbar et al. (1985) revealed that surface sediments on shallow banks have a grain size distribution that generally exceeds 125 μm. This supports our interpretation that the majority of the terrigenous material in U1357B is winnowed from these banks by bottom currents. The size of the material winnowed implies that maximum current velocities in the region are greater than 18-20 cm s$^{-2}$, the minimum velocity required to transport fine sand by intermittent suspension and suspension in poorly sorted glacimarine settings (Singer and Anderson, 1984; McCave and Hall, 2006).

The complete lack of medium to coarse sand in the grainsize distributions of Unit I (<11.4 ka), and from visual observation of the core face, that may represent IBRD may be the consequence that icebergs calved from large ice shelves and ice tongues, such as the RIS and Mertz Glacier Tongue, are advected into the region via the Antarctic Coastal Current but usually lack basal debris. While the lack of IBRD could in part be explained as the consequence of the widespread development of the RIS in the Holocene (McKay et al., 2016), it could also be due to the Adélie and Mertz Banks (and Mertz Glacier Tongue) acting to shield Site U1357 from large icebergs passing over the site, as icebergs would have become grounded on the bathymetric highs and deflected north (Massom et al., 2001; Beaman et al., 2011). Notably, this lack of IBRD further supports a lack of shifting glacial dynamics and calving of sediment-laden icebergs from the smaller local Astrolabe and Zéléé outlet glaciers through this time.

There is a rapid increase in mud content at 4.5 ka coincident with a reduction in both the biogenic and terrigenous MARs, although the terrigenous MAR curve shows higher

[Figure]

**Figure S4** 3.5 kHz seismic profile of the Adélie drift. Top: South (A) to north (A') profile, showing early Holocene (inferred to be ~11-8.2 ka) strata downlapping on top the basement highs (blue reflector and below). This is overlain by onlapping strata (blue reflector and above). Middle and Bottom: West (B) to East (B') profile, showing that in the depocentre of the basin, the pre-8.2 strata has a different geometry to overlying strata (onlapping onto the northern bank of the Dumont D'Urville Trough), suggesting a supply from the south, while overlying strata form a drift deposit that is thickest on the northern flank of the trough, and infers a drift deposit morphology that is aligned with the flow of easterly Antarctic Coastal Current.

accumulation rates than the biogenic MAR curve (Fig. 4). Hence, less material is being advected to the site, and the maximum current strength acting to winnow and advect material from the Adélie Bank into the Dumont d'Urville Trough is reduced. A reduction in maximum current strength could potentially be explained by more extensive sea ice over the site, which would act to reduce wind stress on the ocean surface and thus the maximum strength of the easterly flow, despite enhanced zonal easterly winds that are predicted with a cooler Antarctic climate (Shin et al., 2003; DeConto et al., 2007).

A final consideration is that aeolian contribution of terrigenous material is known to be of importance in Antarctic coastal areas affected by katabatic winds (Atkins and Dunbar, 2009; Chewings et al., 2014). However, windblown sediment is usually well-sorted, and combined with the lack of exposed sediment in Wilkes Land, and the distance of the core site from the coast, input of aeolian sediment into the ocean from melting sea ice is likely to be a relatively minor component of the sediment population relative to the suspended sediment load derived from the local banks and pelagic processes.

**S2.3. Highly-branched isoprenoids (HBI)**

Several recent studies have highlighted the strong potential of the HBI biomarkers along the Antarctic coast as a robust proxy of sea ice extent. Indeed, it has been shown that the di-unsaturated HBI lipid (i.e. diene II or $C_{25:2}$ alkene) is only synthesized in the modern Antarctic waters by sea ice-associated diatoms (Belt et al., 2016; Massé et al., 2011; Smik et al., 2016). The tri-unsaturated HBI lipid (i.e. triene III or $C_{25:3}$) is in contrast strictly produced by open water diatom species, which have been found to be in highest abundance in the marginal ice zone (Smik et al., 2016). Thus, the calculated diene/triene ratio is a reliable tracer to qualitatively estimate the sea ice extent (sea ice vs open water conditions). As previously applied in various Antarctic coastal sediments during different period of time (Etourneau et al., 2013; Collins et al., 2012; Barbara et al., 2010), the diene/triene ratio was successfully used to reconstruct the past sea ice history around Antarctica. In particular, it has been shown that the HBIs were not significantly affected by (i) changes in sources (glacial ice vs sea ice), as the diatoms producing the HBIs strictly grow in relation with sea ice (under or at the edge) (ii) bacterial degradation (Robson and Rowland, 1988) or (iii) rapid sulfurization under anoxic conditions (Sinninghe Damsté et al., 2007). Absolute concentrations of the HBI diene and triene compounds are shown in Fig. S5.

[Figure]

**Figure S5** Absolute concentrations of highly branched isoprenoids (HBIs) measured in U1357B.

**S3. Data-Model mismatch**

Comparison of sea ice data from the Adélie region (presented in this study), with model output for sea ice thickness and extent from TraCE-21k simulations, indicates a clear mismatch between the observational data and model output over the Holocene (Fig. S6). The rapid mid-Holocene increase in sea ice, recorded at Site U1357 and other sites in the Antarctic coastal zone (Fig. S6c, d), is not seen in the model simulations, which instead indicate a sharp decline in sea ice extent and thickness around the Antarctic and in the Adélie region after ca. 5 ka (Fig. S6a, b). The Community Climate

[Figure]

**Figure S6** Comparison of sea ice data from the Adélie region with TraCE-21k simulations a) Antarctic sea ice extent ($10^6$ km$^2$) from TraCE-21k b) Adélie sea ice thickness (66°S, 140°E) from TraCE-21k c) Ratio of the di-unsaturated HBI (C25:2; Diene) and the tri-unsaturated HBI isomer (C25:3; Triene) at Site U1357 d) *Fragilariopsis curta* group relative  abundances from MD03-2601.

System Model (CCSM3) used in the simulations lacks a dynamic ice sheet, instead using a constant prescribed meltwater flux of 1.12 m ka$^{-1}$ from the Antarctic which finishes at 5 ka, which can likely explain the simulated sea-ice decline at ca. 5 ka. Furthermore, it does not incorporate ice-ocean coupling or ice shelf cavities. We have shown that these processes have an important role on sea ice production and thus are required within models to capture the coupled response between ice sheets and the ocean. We note that some models, such as LOVECLIM (Renssen et al., 2010), do simulate a gradual sea-ice increase and cooling trend in the late Holocene, however the timing (gradual) and magnitude (subtle) of sea-ice trends do not match the abrupt and large changes seen in the proxy data.

**Supplementary material references**

Andrews, J.T., Domack, E.W., Cunningham, W.L., et al. (1999) Problems and possible solutions concerning radiocarbon dating of surface marine sediments, Ross Sea, Antarctica. *Quaternary Research*, 52 (2): 206–216. doi:10.1006/qres.1999.2047.

Atkins, C.B. and Dunbar, G.B. (2009) Aeolian sediment flux from sea ice into Southern McMurdo Sound, Antarctica. *Global and Planetary Change*, 69 (3): 133–141. doi:10.1016/j.gloplacha.2009.04.006.

Bahr, A., Jiménez-Espejo, F.J., Kolasinac, N., et al. (2014) Deciphering bottom current velocity and paleoclimate signals from contourite deposits in the Gulf of Cádiz during the last 140 kyr: An inorganic geochemical approach. *Geochemistry, Geophysics, Geosystems*, 15 (8): 3145–3160. doi:10.1002/2014GC005356.

Barbara, L., Crosta, X., Massé, G., et al. (2010) Deglacial environments in eastern Prydz Bay, East Antarctica. *Quaternary Science Reviews*, 29 (19–20): 2731–2740. doi:10.1016/j.quascirev.2010.06.027.

Beaman, R.J., O'Brien, P.E., Post, A.L., et al. (2011) A new high-resolution bathymetry model for the Terre Adélie and George V continental margin, East Antarctica. *Antarctic Science*, 23 (1): 95–103. doi:10.1017/S095410201000074X.

Belt, S.T., Smik, L., Brown, T.A., et al. (2016) Source identification and distribution reveals the potential of the geochemical Antarctic sea ice proxy IPSO25. *Nature Communications*, 7: 12655. doi:10.1038/ncomms12655.

Bentley, M.J., Ocofaigh, C., Anderson, J.B., et al. (2014) A community-based geological reconstruction of Antarctic Ice Sheet deglaciation since the Last Glacial Maximum. *Quaternary Science Reviews*, 100: 1–9. doi:10.1016/j.quascirev.2014.06.025.

Blaauw, M. and Christeny, J.A. (2011) Flexible paleoclimate age-depth models using an autoregressive gamma process. *Bayesian Analysis*, 6 (3): 457–474. doi:10.1214/11-BA618.

Chewings, J.M., Atkins, C.B., Dunbar, G.B., et al. (2014) Aeolian sediment transport and deposition in a modern high-latitude glacial marine environment. *Sedimentology*, 61 (6): 1535–1557. doi:10.1111/sed.12108.

Collins, L.G., Pike, J., Allen, C.S., et al. (2012) High-resolution reconstruction of southwest Atlantic sea-ice and its role in the carbon cycle during marine isotope stages 3 and 2. *Paleoceanography*, 27 (3). doi:10.1029/2011PA002264.

Crosta, X., Denis, D. and Ther, O. (2008) Sea ice seasonality during the Holocene, Adelie Land, East Antarctica. *Marine Micropaleontology*, 66 (3–4): 222–232. doi:10.1016/j.marmicro.2007.10.001.

DeConto, R., Pollard, D. and Harwood, D. (2007) Sea ice feedback and Cenozoic evolution of Antarctic climate and ice sheets. *Paleoceanography*, 22 (3). doi:10.1029/2006PA001350.

Denis, D., Crosta, X., Schmidt, S., et al. (2009a) Holocene glacier and deep water dynamics, Adélie Land region, East Antarctica. *Quaternary Science Reviews*. doi:10.1016/j.quascirev.2008.12.024.

Denis, D., Crosta, X., Schmidt, S., et al. (2009b) Holocene productivity changes off Adélie land (East Antarctica). *Paleoceanography*, 24 (3): 1–12. doi:10.1029/2008PA001689.

Dunbar, R.B., Anderson, J.B., Domack, E.W., et al. (1985) "Oceanographic influences on sedimentation along the Antarctic continental shelf." In *Oceanology of the Antarctic Continental Shelf (eds S. Jacobs)*. pp. 291–312. doi:10.1029/AR043p0291.

Escutia, C., Brinkhuis, H. and Klaus, A. (2011) Site summary. *Proc. IODP /*, 318: 1–74. doi:10.2204/iodp.proc.318.105.2011.

Etourneau, J., Collins, L.G., Willmott, V., et al. (2013) Holocene climate variations in the western Antarctic Peninsula: Evidence for sea ice extent predominantly controlled by changes in insolation and ENSO variability. *Climate of the Past*, 9 (4): 1431–1446. doi:10.5194/cp-9-1431-2013.

Folk, R.L. and Ward, W.C. (1957) Brazos River bar [Texas]; a study in the significance of grain size parameters. *Journal of Sedimentary Research*, 27 (1): 3–26. doi:10.1306/74D70646-2B21-11D7-8648000102C1865D.

Griffith, E.M. and Paytan, A. (2012) Barite in the ocean - occurrence, geochemistry and palaeoceanographic applications. *Sedimentology*, 59 (6): 1817–1835. doi:10.1111/j.1365-3091.2012.01327.x.

Hall, B.L., Henderson, G.M., Baroni, C., et al. (2010) Constant Holocene Southern-Ocean14C reservoir ages and ice-shelf flow rates. *Earth and Planetary Science Letters*, 296 (1–2): 115–123. doi:10.1016/j.epsl.2010.04.054.

Kleinschmidt, G. and Talarico, F. (2000) The Mertz shear zone. *Terra Antarctica Reports*, pp. 109–115.

Massé, G., Belt, S.T., Crosta, X., et al. (2011) Highly branched isoprenoids as proxies for variable sea ice conditions in the Southern Ocean. *Antarctic Science*, 23 (5): 487–498. doi:10.1017/S0954102011000381.

Massom, R.A., Hill, K.L., Lytle, V.I., et al. (2001) Effects of regional fast-ice and iceberg distributions on the behaviour of the Mertz Glacier polynya, East Antarctica. *Annals of Glaciology*,

33: 391–398. doi:10.3189/172756401781818518.

McCave, I.N. and Hall, I.R. (2006) Size sorting in marine muds: Processes, pitfalls, and prospects for paleoflow-speed proxies. *Geochemistry, Geophysics, Geosystems*, 7 (10). doi:10.1029/2006GC001284.

McKay, R., Browne, G., Carter, L., et al. (2009) The stratigraphic signature of the late Cenozoic Antarctic Ice Sheets in the Ross Embayment. *Bulletin of the Geological Society of America*, 121 (11–12): 1537–1561. doi:10.1130/B26540.1.

McKay, R., Golledge, N.R., Maas, S., et al. (2016) Antarctic marine ice-sheet retreat in the Ross Sea during the early Holocene. *Geology*, 44 (1): 7–10. doi:10.1130/G37315.1.

Paytan, A., Mearon, S., Cobb, K., et al. (2002) Origin of marine barite deposits: Sr and S isotope characterization. *Geology*, 30 (8): 747–750. doi:10.1130/0091-7613(2002)030<0747:OOMBDS>2.0.CO;2.

Plewa, K., Meggers, H. and Kasten, S. (2006) Barium in sediments off northwest Africa: A tracer for paleoproductivity or meltwater events? *Paleoceanography*, 21 (2): 1–15. doi:10.1029/2005PA001136.

Powell, R. and Domack, E. (1995) "Glaciomarine processes and sediments." In *Modern Glacial Environments (Eds J. Menzies)*. Butterworth-Heinemann, Oxford.

Presti, M., Barbara, L., Denis, D., et al. (2011) Sediment delivery and depositional patterns off Adélie Land (East Antarctica) in relation to late Quaternary climatic cycles. *Marine Geology*, 284 (1–4): 96–113. doi:10.1016/j.margeo.2011.03.012.

Renssen, H., Goosse, H., Crosta, X., et al. (2010) Early holocene laurentide ice sheet deglaciation causes cooling in the high-latitude southern hemisphere through oceanic teleconnection. *Paleoceanography*, 25 (3): 1–15. doi:10.1029/2009PA001854.

Robson, J.N. and Rowland, S.J. (1988) Biodegradation of highly branched isoprenoid hydrocarbons: A possible explanation of sedimentary abundance. *Organic Geochemistry*, 13 (4–6): 691–695. doi:10.1016/0146-6380(88)90090-3.

Shin, S.-I., Liu, Z., Otto-Bliesner, B.L., et al. (2003) Southern Ocean sea-ice control of the glacial North Atlantic thermohaline circulation. *Geophysical Research Letters*, 30 (2). doi:10.1029/2002GL015513.

Singer, J.K. and Anderson, J.B. (1984) Use of total grain-size distributions to define bed erosion and transport for poorly sorted sediment undergoing simulated bioturbation. *Marine Geology*, 57 (1–4): 335–359. doi:10.1016/0025-3227(84)90204-4.

Sinninghe Damsté, J.S., Rijpstra, W.I.C., Coolen, M.J.L., et al. (2007) Rapid sulfurisation of highly branched isoprenoid (HBI) alkenes in sulfidic Holocene sediments from Ellis Fjord, Antarctica. *Organic Geochemistry*, 38 (1): 128–139. doi:10.1016/j.orggeochem.2006.08.003.

Smik, L., Belt, S.T., Lieser, J.L., et al. (2016) Distributions of highly branched isoprenoid alkenes and other algal lipids in surface waters from East Antarctica: Further insights for biomarker-based paleo sea-ice reconstruction. *Organic Geochemistry*, 95: 71–80. doi:10.1016/j.orggeochem.2016.02.011.

Thomas, D.N. (2011) "Biogeochemistry of Sea Ice." In *Encyclopedia of Snow, Ice and Glaciers (eds Singh V.P., Singh P., Haritashya U.K.)*. pp. 98–102. doi:10.1007/978-90-481-2642-2_639.

Thomas, R.H. and Bentley, C.R. (1978) A model for Holocene retreat of the West Antarctic Ice Sheet. *Quaternary Research*, 10 (2): 150–170. doi:10.1016/0033-5894(78)90098-4.